# $\mu$LO: Compute-Efficient Meta-Generalization of Learned Optimizers

## Abstract

Learned optimizers (LOs) can significantly reduce the wall-clock training time of neural networks, substantially reducing training costs. However, they can struggle to optimize unseen tasks (meta-generalize), especially when training networks much larger than those seen during meta-training. To address this, we derive the Maximal Update Parametrization ($\mu$P) for two popular learned optimizer architectures and propose a simple meta-training recipe for $\mu$-parameterized LOs ($\mu$LOs). Our empirical evaluation demonstrates that LOs meta-trained with our recipe substantially improve meta-generalization to wider unseen tasks when compared to LOs trained under standard parametrization (e.g., as they are trained in existing work). When applying our $\mu$LOs, each trained for less than 250 GPU-hours, to large-width models we are often able to match or exceed the performance of pre-trained VeLO, the most performant publicly available learned optimizer, meta-trained with 4000 TPU-months of compute. We also empirically observe that learned optimizers trained with our $\mu$LO recipe also exhibit substantially improved meta-generalization to deeper networks ($5\times$ meta-training) and remarkable generalization to much longer training horizons ($25\times$ meta-training).

## 1 Introduction

Deep learning (Goodfellow et al., 2016) has enabled a great number of breakthroughs (Brown et al., 2020; Brooks et al., 2024; Radford et al., 2021; Alayrac et al., 2022; Kirillov et al., 2023; Rombach et al., 2022; Oquab et al., 2023). Its success can, in part, be attributed to its ability to learn effective representations for downstream tasks. Notably, this resulted in the abandonment of a number of heuristics (e.g., hand-designed features in computer vision (Dalal and Triggs, 2005; Lowe, 2004)) in favor of end-to-end learned features. However, one aspect of the modern deep-learning pipeline remains hand-designed: gradient-based optimizers. While popular optimizers such as Adam or SGD provably converge to a local minimum in non-convex settings (Kingma and Ba, 2017; Li et al., 2023; Robbins, 1951), there is no reason to expect these hand-designed optimizers reach the global optimum at the optimal rate for a given problem. Given the lack of guaranteed optimality and the clear strength of data-driven methods, it is natural to turn towards data-driven solutions for improving the optimization of neural networks.

To improve hand-designed optimizers, Andrychowicz et al. (2016); Wichrowska et al. (2017); Metz et al. (2019; 2022a) replaced them with small neural networks called learned optimizers (LOs). Metz et al. (2022b) showed that scaling up the training of such optimizers can significantly improve wall-clock training speeds and supersede existing hand-designed optimizers. However, LOs have limitations in *meta-generalization* – optimizing new problems. For example, despite training for 4000 TPU months, VeLO (Metz et al., 2022b) is known to (1) generalize poorly to longer optimization problems (e.g., more steps) than those seen during meta-training and (2) have difficulty optimizing models much larger than those seen during meta-training. Given the high cost of meta-training LOs (e.g., when meta-training, a *single training example* is analogous to training a neural network for many steps), it is essential to be able to train learned optimizers on small tasks and generalize to larger ones. Harrison et al. (2022) explore preconditioning methods to improve the generalization from shorter to longer optimization problems (e.g., ones with more steps). However, no works have tackled the meta-generalization of LOs to wider models in a principled way.

To address the meta-generalization problem of LOs, we recognize that this problem can be reformulated as *zero-shot hyperparameter transfer* (Yang et al., 2022). The latter involves selecting optimal hyperparameters of hand-designed optimizers for training very large networks (that one

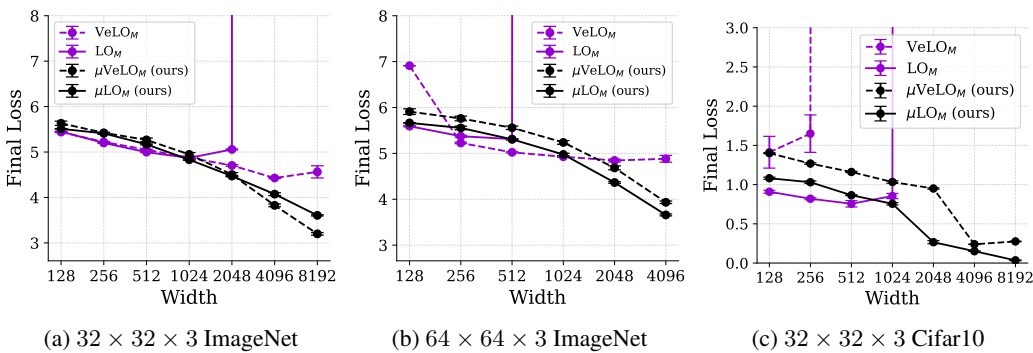

(a) $32 \times 32 \times 3$ ImageNet    (b) $64 \times 64 \times 3$ ImageNet    (c) $32 \times 32 \times 3$ Cifar10

Figure 1: **Generalization beyond meta-training widths is severely limited without our approach.** We report the final loss after 1000 steps (e.g., the inner problem length used when meta-training) for models of different widths. Each point is the average final training loss over 5 seeds with standard error bars. We observe that both $\mu$LOs consistently obtain lower loss values as the tasks become wider. In contrast, their SP LO counterparts either diverge before reaching 1000 steps on the wider tasks or make little progress as width is increase.

cannot afford to tune directly) by transferring those tuned on smaller versions of the model. Under the standard parametrization (SP)[1], the optimal hyperparameters of an optimizer used for a small model do not generalize well to larger versions of the model. However, when a small model is tuned using the Maximal Update Parametrization ($\mu$P), and its larger counterparts are also initialized with $\mu$P, the small and large models share optimal hyperparameters (Yang et al., 2022). Given the appealing connection between zero-shot hyperparameter transfer in hand-crafted optimizers and meta-generalization in LOs, we ask the following questions: *Can learned optimizers be meta-trained under $\mu$P? How would the resulting optimizers perform on wider unseen tasks?* We seek to answer these questions in the following study. Specifically, we consider the recent LO architectures (Metz et al., 2022a;b) and demonstrate that $\mu$P can be adapted to these optimizers leading to our $\mu$LO optimizers. We subsequently conduct an empirical evaluation that reveals the power of our $\mu$LOs and their advantages for scaling learned optimizers.

Our contributions can be summarized as follows:

- We derive $\mu$-parameterization for two popular learned optimizer architectures (VeLO and small_fc_lopt) and propose a training recipe for $\mu$LOs.

- We demonstrate that $\mu$LOs meta-trained with our recipe significantly improve generalization to wider networks when compared to their SP counterparts and several strong baselines and that, for wider counterparts of the meta-training tasks, they outperform VeLO (meta-trained with 4000 TPU-months of compute).

- We demonstrate empirically that $\mu$LOs meta-trained with our recipe show improved generalization to deeper networks ($5\times$ meta-training) when compared to their SP counterparts.

- We demonstrate empirically that $\mu$LOs meta-trained with our recipe significantly improve generalization to longer training horizons ($25\times$ meta-training) when compared to their SP counterparts.

Our results show that $\mu$LOs significantly improve learned optimizer generalization without increasing meta-training costs. This constitutes a noteworthy improvement in the scalability of meta-training LOs.

## 2   RELATED WORK

**Learned optimization.**   While research on learned optimizers (LOs) spans several decades (Schmidhuber, 1992; Thrun and Pratt, 2012; Chen et al., 2022; Amos, 2022), our work is primarily related to

---

[1]When we refer to SP, we follow the same meaning as Yang et al. (2022). That is, we consider SP to designate a parameterization that does not admit HP transfer. However, we note that recent work (Everett et al., 2024) shows hyperparameter transfer is possible in SP under certain alignment assumptions.

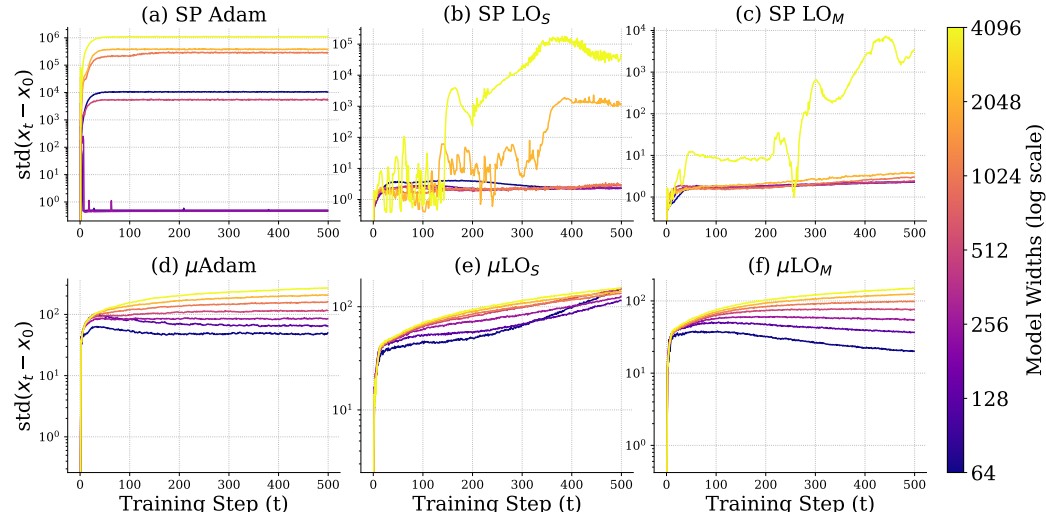

Figure 2: **Layer 2 pre-activations behave harmoniously in $\mu$P for $\mu$LOs and $\mu$Adam alike.** We report the evolution of coordinate-wise standard deviation of the difference between the initial ($t = 0$) and $t$-th second-layer pre-activations of an MLP during training for the first 500 steps of a single run (the remaining layers behave similarly, see Sec. I). We observe that all models parameterized in $\mu$P enjoy stable coordinates across widths, while the pre-activations of larger-width models in SP blow up after a number of training steps.

the recent meta-learning approaches utilizing efficient per-parameter optimizer architectures of Metz et al. (2022a). Unlike prior work (Andrychowicz et al., 2016; Wichrowska et al., 2017; Chen et al., 2020), which computes meta-gradients (the gradients of the learned optimizer) using backpropagation, Metz et al. (2022a) use Persistent Evolutionary Strategies (PES) (Vicol et al., 2021), a truncated variant of evolutionary strategies (ES) (Buckman et al., 2018; Nesterov and Spokoiny, 2017; Parmas et al., 2018). ES improves meta-training of LOs by having more stable meta-gradient estimates compared to backpropagation through time, especially for longer sequences (i.e. long parameter update unrolls inherent in meta-training) (Metz et al., 2019). PES and most recently ES-Single (Vicol, 2023) are more efficient and accurate variants of ES, among which PES is more well-established in practice making it a favourable approach to meta-training.

**Generalization in LOs.** One of the critical issues in LOs is generalization in the three main aspects (Chen et al., 2022; Amos, 2022): (1) optimize novel tasks (often referred to as *meta-generalization*); (2) optimize for more iterations than the maximum unroll length used in meta-training; (3) avoid overfitting on the training set. Among these, (3) has been extensively addressed using different approaches, such as meta-training on the validation set objective (Metz et al., 2019), adding extra-regularization terms (Harrison et al., 2022), parameterizing LOs as hyperparameter controllers (Almeida et al., 2021) and introducing flatness-aware regularizations (Yang et al., 2023). The regularization terms (Harrison et al., 2022; Yang et al., 2023) often alleviate issue (2) as a byproduct. However, meta-generalization (1) has remained a more difficult problem. One approach to tackle this problem is to meta-train LOs on thousands of tasks (Metz et al., 2022b). However, this approach is extremely expensive and does not address the issue in a principled way leading to poor meta-generalization in some cases, e.g. when the optimization task includes much larger networks. Alternatively, Premont-Schwarz et al. (2022) introduced Loss-Guarded L2O (LGL2O) that switches to Adam/SGD if the LO starts to diverge improving meta-generalization. However, this approach needs tuning Adam/SGD and requires additional computation (e.g. for loss check) limiting (or completely diminishing in some cases) the benefits of the LO. In this work, we study aspects (1) and (2) of LO generalization, demonstrating how existing SP LOs generalize poorly across these dimensions and showing how one can apply $\mu$P to learned optimizers to substantially improve generalization in both these aspects.

**Maximal Update Parametrization.** First proposed by Yang and Hu (2021), the Maximal Update Parametrization is the unique stable abc-Parametrization where every layer learns features. The

parametrization was derived for adaptive optimizers by Yang and Littwin (2023) and was applied by Yang et al. (2022) to enable zero-shot hyperparameter transfer, constituting the first practical application of the tensor programs series of papers. Earlier works in the *tensor programs series* build the mathematical foundation that led to the discovery of $\mu$P. Yang (2019) shows that many modern neural networks with randomly initialized weights and biases are Gaussian Processes, providing a language, called Netsor, to formalize neural network computations. Yang (2020a) focuses on neural tangent kernels (NTK), proving that as a randomly initialized network's width tends to infinity, its NTK converges to a deterministic limit. Yang (2020b) shows that randomly initialized network's pre-activations become independent of its weights when its width tends to infinity. Most recently, in tensor programs VI, Yang et al. (2024) propose Depth-$\mu$P, a parameterization allowing for hyperparameter transfer in infinitely deep networks. However, Depth-$\mu$P is only valid for residual networks with a block depth of 1, making it unusable for most practical architectures (e.g., transformers, resnets, etc.). For these reasons, we do not study Depth-$\mu$P herein. Building on the latest works studying width $\mu$P (Yang and Littwin, 2023; Yang et al., 2022), in this work, we show that $\mu$P can be extended to the case of learned optimizers and empirically evaluate its benefits in this setting.

## 3 METHOD

### 3.1 BACKGROUND

A standard approach to learning optimizers (Metz et al., 2022a) is to solve the following meta-learning problem:

$$\min_{\phi} \; \mathbb{E}_{(\mathcal{D}, \boldsymbol{w}_0) \sim \mathcal{T}} \left[ \mathbb{E}_{(X,Y) \sim \mathcal{D}} \left[ \frac{1}{T} \sum_{t=0}^{T-1} \mathcal{L}(X, Y; f_{\phi}(\boldsymbol{u}_t), \boldsymbol{w}_t) \right] \right], \tag{1}$$

where $\mathcal{T}$ is a distribution over optimization tasks defined as pairs of dataset $\mathcal{D}$ and initial weights $\boldsymbol{w}_0$ associated with a particular neural architecture (we refer to this network as the *optimizee*), $\phi$ represents the weights of the learned optimizer, $f_{\phi}$, that takes gradient-based features $\boldsymbol{u}_t$ as input. Finally, $\mathcal{L}$ is the loss function used to train the optimizee. $T$ is the length of the unroll which we write as a fixed quantity for simplicity. In our experiments, during meta-optimization, $T$ is varied according to a truncation schedule (Metz et al., 2022a). A clear goal of the learned optimization community is not only learning to solve optimization problems over $\mathcal{T}$, but also to apply the learned optimizer, $f_{\phi}$, more generally to unobserved datasets and architectures. This *transfer* to new tasks is referred to as meta-generalization. This problem can be seen as a generalization of the zero-shot hyperparameter transfer problem considered in Yang et al. (2022); for instance, when the optimizer is a hand-designed method such as SGD or Adam and $\phi$ represents optimization hyper-parameters such as the learning rate.

Gradient descent is a standard approach to solving equation 1. However, estimating the meta-gradients via backpropagation for very long unrolls is known to be noisy (Metz et al., 2019). Instead, gradients are estimated using evolution strategies (Buckman et al., 2018; Nesterov and Spokoiny, 2017; Parmas et al., 2018). Evolution strategies work by sampling perturbations to the LO's weights (similar to SPSA (Spall, 2000)), unrolling an optimization trajectory for each perturbation, and estimating gradients with respect to evaluations of the meta-objective (usually the loss of the network being optimized, see eq. 1). In contrast to ES, which estimates one gradient per full unroll, PES (Vicol et al., 2021) allows estimating unbiased gradients at many points (called truncations) during the full unroll. This allows updating the optimizer's parameters more often during meta-training. We use PES to estimate meta-gradients in our experiments.

Learned optimizer features $\boldsymbol{u}_t$ are constructed based on momentum, second-order momentum, and adafactor values as in (Metz et al., 2022a), with the full list of features described in the (Table 6 of the Appendix). In our experiments, the architectures of our $f_{\phi}$ are similar to **small_fc_lopt** of Metz et al. (2022a) and **VeLO** of Metz et al. (2022b) except that their dimensions differ slightly (see sec. C for details). $f_{\phi}$ has three outputs $m$, $d$, and $\alpha$, the magnitude, scale, and learning rate of the update respectively. For **small_fc_lopt**, $\alpha_w = 1$ always, $\alpha_w$ is produced by the tensor-level LSTM for VeLO. The standard LO update is given as

$$w_t = w_{t-1} - \alpha_w \lambda_1 d_{\phi} \exp\left(\lambda_2 m_{\phi}\right), \tag{2}$$

where $\lambda_1$ and $\lambda_2$ are constant values of 0.001 to bias initial step sizes towards being small.

### 3.2 $\mu$-PARAMETRIZATION FOR LEARNED OPTIMIZERS

Parameterizing an optimizee neural network in $\mu$P requires special handling of the initialization variance, pre-activation multipliers, and optimizer update for each weight matrix $w \in \mathbb{R}^{n \times m}$ in the network. Specifically, these quantities will depend on the functional form of the optimizer and the dependence of $n$ (FAN_OUT) and $m$ (FAN_IN) on width. We will refer to weight matrices in a network of width $h$ as hidden layers if $\Theta(n) = \Theta(m) = \Theta(h)$, as output layers if $\Theta(n) = 1, \Theta(m) = \Theta(h)$, and as input layers if $\Theta(n) = \Theta(h), \Theta(m) = \Theta(h)$.

Consider a model *to be optimized* $g_w$ with weights in layers $l$ denoted $w_l$. We apply and construct $\mu$LOs as follows.

**Initialization-$\mu$.** $w_l$ which are hidden and input layers have their weights initialized as $\mathcal{N}(0, \frac{1}{\sqrt{\text{FAN\_IN}}})$. While output layers have their weights initialized as $\mathcal{N}(0, 1)$.

**Multipliers-$\mu$.** Output layer pre-activations are multiplied by $\frac{1}{\text{FAN\_IN}}$ during the forward pass.

**Updates-$\mu$.** The update by $f_\phi$ on the parameters of $g_w$, at both meta-training and evaluation is modified as follows:

$$w_t = \begin{cases} w_{t-1}^i - \frac{1}{\text{FAN\_IN}} \cdot \left( \alpha_w \lambda_1 d_\phi^i \exp\left( \lambda_2 m_\phi^i \right) \right) & \text{if } w^i \text{ is part of a hidden layer} \\ w_{t-1}^i - \alpha_w \lambda_1 d_\phi^i \exp\left( \lambda_2 m_\phi^i \right) & \text{otherwise.} \end{cases} \tag{3}$$

We now show that this can lead to a maximal update Parametrization, following the analysis of (Yang et al., 2022, Appendix J.2.1) which studies the initial optimization step. For our analysis, we consider a simplified input set for $f_\phi$ which takes as input only the gradient while producing an update for each layer. Note that this analysis extends naturally to other first-order quantities.

**Proposition 1.** *Assume that the LO $f_\phi$ is continuous around 0. Then, if $f_\phi(0) \neq 0$, the update, initialization, and pre-activation multiplier above is necessary to obtain a Maximal Update Parametrization.*

### 3.3 $\mu$LO META-TRAINING RECIPE

$\mu$P for hand-designed optimizers involves tuning the optimizer on a small width version of the target architecture and transferring the hyperparameters to the larger width target model (Yang et al., 2022). While $\mu$-transfer makes hyperparameter search for large models tractable, it has the following limitations: (1) the smaller scale hyperparameter search suffers from increased complexity as it requires sweeping various multipliers in addition to the standard hyperparameters, (2) tuning the hyperparameters on too small of a model may result in sub-optimal hyperparameters for the largest models, (3) Yang et al. (2022) recommend repeating the procedure for every new task/dataset. Meta-training flexible $\mu$-parametrized learned optimizers can address these limitations. Due to their flexible functional forms (as opposed to just a learning rate hyperparameter), $\mu$LOs can learn to optimize networks in $\mu$P without tuning multipliers (we set all multipliers to 1 in our experiments). Therefore, by training our $\mu$LOs with fixed multipliers on *multiple tasks* that are large enough to ad-

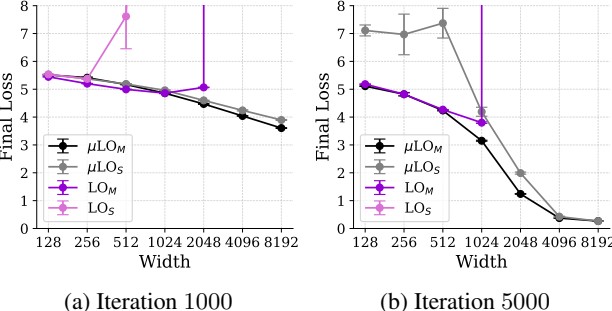

(a) Iteration 1000         (b) Iteration 5000

Figure 3: $\mu$**LO$_S$ underperforms $\mu$LO$_M$ as width and training steps increase.** Each point is the average training loss over 5 seeds at iterations 1000 (a) or 5000 (b). Error bars report standard error.

mit strong transfer but still tractable and reusing them on new tasks, we address (1), (2), and (3) by amortizing the tuning cost during the optimizer meta-training stage. However, it should be noted that while the $\mu$LO framework allows for meta-generalization to unseen new tasks (unlike $\mu$-transfer), a $\mu$LO that relies on meta-generalization for transfer to new tasks should expect to be outperformed by the $\mu$LO that also meta-trains on a small version of that task.

Table 1: **Meta-training configurations of LOs and baselines in our empirical evaluation.**

| Identifier | Type | Architecture | Optimizee Par. | Meta-Training / Tuning Task(s) |
|---|---|---|---|---|
| $\mu$LO$_S$ | Ours | small_fc_lopt (Metz et al., 2022a) | $\mu$LO Sec. 3.2 | ImageNet classification, 3-Layer MLP, width $\in \{128\}$ |
| $\mu$LO$_M$ | Ours | small_fc_lopt (Metz et al., 2022a) | $\mu$LO Sec. 3.2 | ImageNet classification, 3-Layer MLP, width $\in \{128, 512, 1024\}$ |
| $\mu$VeLO$_M$ | Ours | VeLO (Metz et al., 2022b) | $\mu$LO Sec. 3.2 | ImageNet classification, 3-Layer MLP, width $\in \{128, 512, 1024\}$ |
| LO$_S$ | LO Baseline | small_fc_lopt (Metz et al., 2022a) | SP | ImageNet classification, 3-Layer MLP, width $\in \{128\}$ |
| LO$_M$ | LO Baseline | small_fc_lopt (Metz et al., 2022a) | SP | ImageNet classification, 3-Layer MLP, width $\in \{128, 512, 1024\}$ |
| VeLO$_M$ | LO Baseline | VeLO (Metz et al., 2022b) | SP | ImageNet classification, 3-Layer MLP, width $\in \{128, 512, 1024\}$ |
| VeLO-4000 | Oracle LO Baseline | VeLO (Metz et al., 2022b) | SP | We refer the reader to (Metz et al., 2022b, Appendix C.2) |
| $\mu$Adam | Baseline | – | $\mu$P Adam | ImageNet classification, 3-Layer MLP, width $\in \{1024\}$ |
| AdamW | Baseline | – | SP | ImageNet classification, 3-Layer MLP, width $\in \{1024\}$ |

To verify the effectiveness of this multi-task strategy for learned optimizers, we compare $\mu$LO$_S$, trained on a single small task (see Tab. 1), to $\mu$LO$_M$, trained on 3 small tasks of the different width (see Tab. 1), in figure 3. When training for 1000 steps (meta-training unroll length), we observe that $\mu$LO$_M$ outperforms $\mu$LO$_S$ as the width of the model is increased (Fig. 3 (a)). Moreover, we observe that there is a discrepancy in performance between both models after 5000 steps (Fig. 3 (b)), showing that meta-training with multiple tasks of different widths has benefits for generalization to longer unrolls in addition to improved generalization to larger optimizees. Given the improved generalization of $\mu$LO$_M$ compared to $\mu$LO$_S$, we adopt the multiple-width single-task meta-training recipe as part of our method. Subsequent experiments (e.g., figures 1 and 4) will show that it is also effective for meta-training $\mu$VeLO.

## 4    EMPIRICAL EVALUATION

We use a suite of optimization tasks of varying width to evaluate meta-generalization properties of our $\mu$LOs vs tuned $\mu$Adam (Yang et al., 2022), SP AdamW Loshchilov and Hutter (2019), and baseline SP LOs. We also include pre-trained VeLO (Metz et al., 2022b) as an oracle which we denote as VeLO-4000. Meta-trained for 4000 TPUv4 months, it is the strongest publicly available pre-trained learned optimizer. We focus on evaluating generalization to wider networks, however, we also establish the generalization properties of $\mu$LOs to longer training horizons and deeper networks. Please note that while $\mu$LOs inherit the theoretical properties of $\mu$P for width scaling, our findings with respect to longer training and deeper networks are purely empirical.

**Baseline LOs and $\mu$LOs.** The meta-training configuration of each learned optimizer is summarized in Table 1. Each learned optimizer (ours and the baselines) in our empirical evaluation is meta-trained using the multiple-width single-task meta-training recipe proposed in section 3.3. The baseline sheds light on whether simply varying the SP optimizee width during meta-training is enough to achieve generalization of the LO to wider networks in SP. During meta-training, we set the inner problem length to be 1000 iterations. Therefore, any optimization beyond this length is considered out-of-distribution. For all meta-training and hyperparameter tuning details, including ablation experiments, see section C of the appendix.

$\mu$**Adam** $\mu$Adam is a strong hand-designed $\mu$P baseline. It follows the Yang et al. (2022) Adam $\mu$-parametrization and does not use weight decay as this is incompatible with $\mu$P. It is tuned on the largest meta-training task seen by our learned optimizers (Table 1). We tune the learning rate and three multipliers: input multiplier, output multiplier, and the hidden learning rate multiplier. These multipliers correspond to adding a tunable constant to the pre-activation multiplier for input weights, the pre-activation multiplier for output weights, and the Adam LR for hidden weights. More details about the grid search over 500 configurations are provided in Section B.1 of the appendix.

**AdamW** AdamW (Loshchilov and Hutter, 2019) is a strong hand-designed SP baseline. It is tuned on the largest meta-training task seen by our learned optimizers (Table 1). We tune the learning rate, $\beta_1$,$\beta_1$, and the weight decay. More details about the grid search over 500 configurations are provided in Section B.1 of the appendix.

**Pre-trained VeLO (VeLO-4000).** VeLO (Metz et al., 2022b) is a learned optimizer that was meta-trained on a curriculum of progressively more expensive meta-training tasks for a total of 4000 TPU months. These tasks include but are not limited to image classification with MLPs, ViTs, ConvNets, and ResNets; compression with MLP auto-encoders; generative modeling with VAEs; and language modeling with transformers and recurrent neural networks. During meta-training, VeLO-4000 unrolls inner problems for up to 20k steps ($20\times$ ours); the final model was then fine-tuned on tasks with up to 200k steps of optimization. VeLO-4000, therefore, represents the strongest baseline in our empirical evaluation and we consider it to be an oracle.

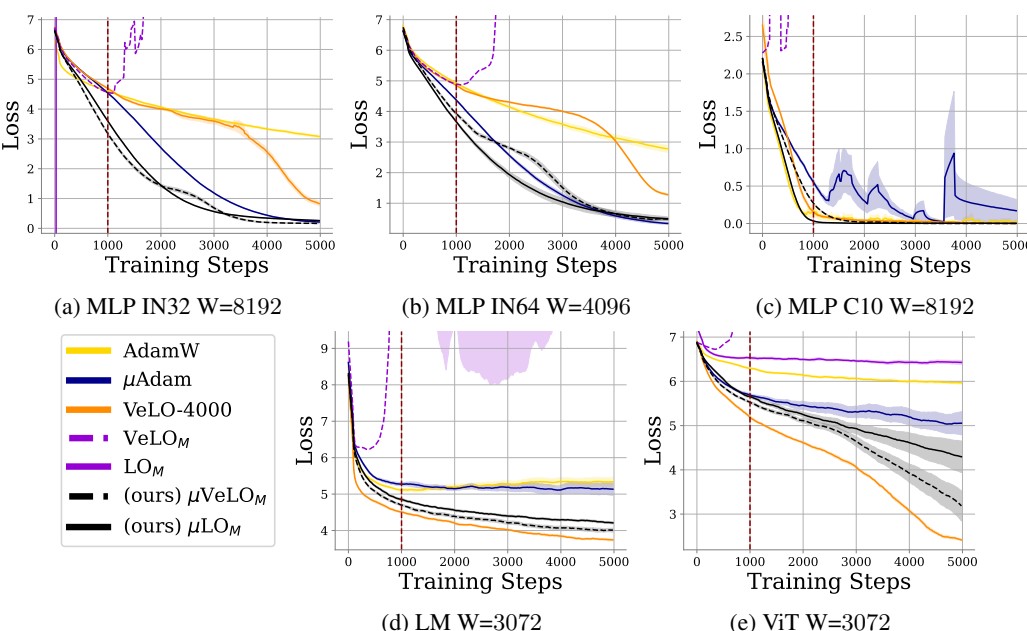

Figure 4: **Evaluating generalization to wider networks for different tasks.** Tasks Our optimizers are meta-trained for 1000 inner steps (dotted red line), therefore, any optimization beyond 1000 steps is considered out-of-distribution. We plot average training loss over 5 seeds with standard error bars. We observe that $\mu$LO$_M$ and $\mu$VeLO$_M$ generalize smoothly to longer unrolls and all unseen tasks, unlike their SP counterparts which diverge or fail to make progress. $\mu$LOs even surpass or match the performance of VeLO in subfigures (a), (b), and (c)). Moreover, they also substantially best the well-tuned hand-designed baselines on LM and ViT tasks (subfigures (d) and (e)) and best or match the best performing hand-designed optimizer in subfigures (a),(b), and (c).

**Is VeLO-4000 a fair baseline?** While we believe the comparison is important given the relevance of our results to scaling up LOs, we highlight that the comparison will unfairly advantage VeLO-4000 as all tasks in our suite fall within its meta-training distribution and VeLO-4000 was meta-trained on inner unroll horizons well beyond those we evaluate. Thus, when comparing our LOs to VeLO-4000, it is important to keep in mind that ours are meta-trained with only $0.004\%$ of VeLO-4000's compute budget.

**Evaluation tasks.** Our evaluation suite includes 35 tasks spanning image classification (CIFAR-10, ImageNet) using MLPs and Vision Transformers (ViTs) (Dosovitskiy et al., 2020) and autoregressive language modeling with a decoder-only transformer on LM1B (Chelba et al., 2013). To create the tasks, we further vary image size (for image classification), width, and depth of the optimizee network, and the number of optimization steps. See Table 7 of the appendix for an extended description of all the tasks.

### 4.1 RESULTS

In the following sections, we first (Sec. 4.1.1) present results empirically verifying the pre-activation stability of our $\mu$LOs. Subsequently, we present the results of our main empirical evaluation of meta-generalization to wider networks (Sec. 4.1.1), a study of $\mu$LOs generalization to deeper networks (Sec. 4.1.3), and a study of $\mu$LOs generalization to longer training horizons (Sec. 4.1.4). All of our figures reporting training loss show the average loss across 5 random seeds. The error bars in these plots report the standard error. Each seed corresponds to a different ordering of training data and a different initialization of the optimizee.

#### 4.1.1 EVALUATING PRE-ACTIVATION STABILITY

We now verify that desiderata J.1 of Yang et al. (2022) is satisfied empirically. In Figure 2, we report the evolution of the coordinate-wise standard deviation of the difference between initial (t=0) and current (t) second-layer pre-activations of an MLP during the first 500 steps of training for a single trial. We observe that all models parameterized in $\mu$P enjoy stable coordinates across widths, while the pre-activations of the larger models in SP blow up after a number of training steps. Notably, SP

Adam's pre-activations blow up immediately, while $LO_S$ and $LO_M$ take longer to blow up and have a more erratic pattern; we hypothesize that this is a side effect of meta-training where the optimizers may learn to keep pre-activations small by rescaling updates. Section I of the appendix contains similar plots for the remaining layers of the MLP which show similar trends.

*In summary, we find, empirically, that pre-activations of µLOs and µAdam are similarly stable across widths, while the activations of SP Adam and SP LOs both blow up but behave qualitatively differently.*

Table 2: **In-distribution and out-of-distribution average performance of optimizers.** We report the average rank of different optimizers across the five tasks in our suite. We evaluate in-distribution at a Base width of 1024 as this is the width used to tune the hand-designed baselines. We also evaluate out-of-distribution widths: Large (2048) and XL (largest size for each task see Tab.7 of the appendix). We bold the strongest, underline the second strongest, and italicize the third strongest average rank in each column. We do not bold entries of VeLO-4000 as it is reported only for reference since it is not a fair comparison. We observe that, across all iterations, when compared to fair baselines, $µLO_M$ obtains the best rank for all settings except for the XL task at 5000 iterations, where it is only bested by $µVeLO$.

| Optimizer | Loss at 1k steps | | | Loss at 3k steps | | | Loss at 5k steps | | |
| --- | --- | --- | --- | --- | --- | --- | --- | --- | --- |
| | ID (Base) | OoD (Large) | OoD (XL) | ID (Base) | OoD (Large) | OoD (XL) | ID (Base) | OoD (Large) | OoD (XL) |
| AdamW | 3.40 | 3.20 | 4.60 | 3.60 | 3.80 | 4.80 | *4.00* | 4.40 | 4.80 |
| µAdam | *4.40* | 4.20 | *4.00* | *3.60* | *3.60* | *3.40* | 3.60 | *3.60* | *3.20* |
| VeLO$_M$ | 5.00 | 5.80 | 5.60 | 6.80 | 6.40 | 7.00 | 7.00 | 7.00 | 6.80 |
| LO$_M$ | 5.20 | 6.60 | 6.80 | 5.20 | 6.60 | 6.00 | 5.60 | 6.00 | 6.20 |
| µVeLO$_M$ (ours) | 4.40 | *3.40* | 2.20 | 3.60 | 2.60 | 2.20 | 4.00 | 2.80 | **2.00** |
| µLO$_M$ (ours) | **2.80** | **1.80** | **2.00** | **2.40** | **2.00** | **1.80** | **2.20** | **2.00** | 2.80 |
| VeLO-4000 | 2.80 | 3.00 | 2.80 | 2.80 | 3.00 | 2.80 | 1.60 | 2.20 | 2.20 |

### 4.1.2 META-GENERALIZATION TO WIDER NETWORKS

Given our goal of improving LO generalization to unseen wider tasks, the bulk of our empirical evaluation is presented in this section. Specifically, we evaluate the behavior of µLOs as the width of tasks increases well beyond what was seen during meta-training. To accomplish this, we fix the depth of each task and vary the width (see Table 7 for a full list of tasks), leading to a testbed of 32 different tasks. We then train each task using the baselines and µ-optimizers outlined in section 4 for 5000 steps for 5 different random seeds. This involves training 1120 different neural networks. To make the results easily digestible, we summarize them by width and final performance in Figure 4 and by average optimizer rank in Table 2. We also highlight the smooth training dynamics of our optimizers at the largest widths in figure 4.

**Performance measured by final loss as a function of width** Figure 1 compares the training loss after 1000 steps of SP learned optimizers to µ-parameterized learned optimizers for different widths. This is shown in three subfigures for three MLP image classification tasks: (a) Imagenet $32 \times 32 \times 3$ (IN32), (b) Imagenet $64 \times 64 \times 3$ (IN64), and (c) Cifar-10 $32 \times 32 \times 3$ (C10). Subfigure (a) shows the performance of learned optimizers on larger versions of the meta-training tasks. We observe that the µLOs achieve lower final training loss as the width of the task is increased. In contrast, $LO_M$ diverges for widths larger than 2048 and VeLO$_M$ fails to substantially decrease the loss at larger widths, falling behind the µLOs. Subfigure (b) evaluates our µLOs of larger ImageNet images (e.g., when the input width is larger). Similarly, we observe smooth improvements in the loss as the optimizee width increases for µLOs, while their SP counterparts either diverge at width 512 ($LO_M$) or fail to substantially improve the loss beyond width 1024 (VeLO$_M$). Finally, Subfigure (c) shows the performance of our µLOs on Cifar-10 (smaller output width) as the width of the model is increased. Similarly, we observe smooth improvements in the loss as the width increases for µLOs, while their SP counterparts either diverge immediately at small widths (VeLO$_M$) or diverge by width 1024 ($LO_M$).

**Performance measured by average optimizer rank** Table 2 reports the average rank of different optimizers on in-distribution width tasks (Base, width 1024) and out-of-distribution width tasks (Large (width 2048) and XL (maximum width)). Each entry of the table corresponds to the optimizer's average rank (within the 7 optimizers evaluated) over the 5 tasks in our suite: Cifar 10 MLP image classification, ImageNet 32 MLP image classification, ImageNet 64 MLP image classification, ImageNet 32 ViT image classification, and LM1B transformer language modelling. The optimizers are ranked by their training loss at the given iteration. We report average ranks for 1000 iterations (inner-problem length), 3000 iterations, and 5000 iterations. We bold the strongest, underline the

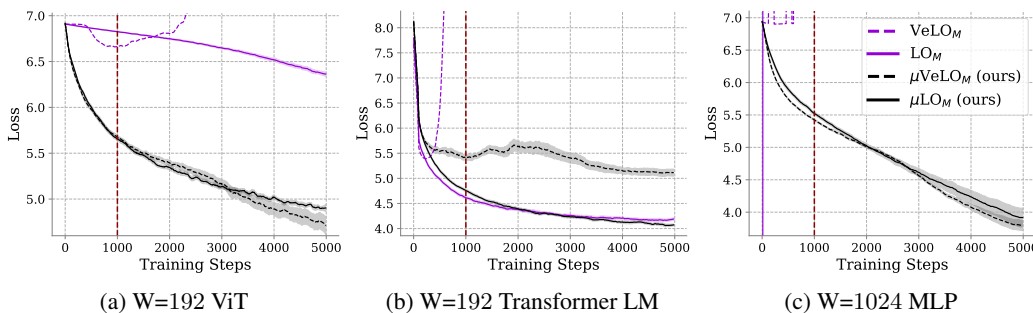

(a) W=192 ViT   (b) W=192 Transformer LM   (c) W=1024 MLP

Figure 5: **Evaluating generalization capabilities of $\mu$LOs to deeper networks**. The figures report the performance of learned optimizers for training depth-16 ViTs for image classification, Transformers for language modeling, and MLPs for image classification. We plot average training loss over 5 seeds with standard error bars. In each case, $\mu$LOs show improved generalization and performance when compared to their SP counterparts.

second strongest, and italicize the third strongest average rank in each column. We do not bold entries of VeLO-4000 as it is reported only for reference since it is not a fair comparison. We observe that, across all iterations, when compared to fair baselines, $\mu$LO$_M$ obtains the best rank for all settings except for the XL task at 5000 iterations, where it is only bested by $\mu$VeLO. When only looking at the out-of-distribution Large and XL tasks, we observe that $\mu$LO$_M$ and $\mu$VeLO$_M$ dominate the first two spots of the optimizer podium in all cases except one. For the Large task at 1000 steps, $\mu$VeLO$_M$ is bested by AdamW. When comparing our $\mu$LOs to VeLO-4000, we observe that at least one of $\mu$LO$_M$ and $\mu$VeLO$_M$ bests VeLO-4000 on all tasks except for the large task at 5000 iterations. This is remarkable as our $\mu$LOs are trained on many orders of magnitude less compute than VeLO-4000. These results demonstrate that meta-training LOs using our recipe yields substantial improvements in meta-generalization (across various tasks and widths) over LOs from previous work and strong hand-designed baselines.

**Training dynamics at the largest widths**   Figure 4 reports the training curves of different optimizers on the largest width tasks in our suite. Despite training for $5\times$ longer than the maximum meta-training unroll length, our $\mu$LOs are capable of smoothly decreasing the loss for the largest out-of-distribution tasks in our suite. In contrast, the strong SP LO baselines diverge by 1000 steps (subfigures (a),(b),(c),(d)), or fail to decrease the training loss (subfigure (e)). Our $\mu$LOs also substantially best the well-tuned hand-designed baselines on LM and ViT tasks (subfigures (d) and (e)) and best or match the best performing hand-designed optimizer in subfigures (a),(b), and (c). Notably in figure (c), our $\mu$LOs can even generalize beyond the tuning/meta-training widow to tasks with a smaller output layer while $\mu$Adam suffers from instability in this case. When comparing with VeLO-4000, we observe that our $\mu$LOs substantially outperform VeLO in subfigures (a),(b), and $\mu$LO$_M$ outperforms VeLO-4000 in subfigure (c). In contrast, VeLO-4000 outperforms our $\mu$LOs on transformer language modeling and ViT image classification, the most out-of-distribution tasks for them. These findings show that $\mu$LOs can outperform VeLO-4000 on larger in-distribution tasks, suggesting that scaling meta-training in SP (e.g., as done for VeLO) may not be sufficient to achieve strong meta-generalization to the largest tasks, but that meta-training in $\mu$P could be.

*In summary, the results in Fig. 1,Tab. 2 and Fig.4 demonstrate that our $\mu$LO meta-training recipe represents a considerable advancement to low-cost meta-generalization for learned optimizers. The technique is shown to be a substantial improvement over previous work.*

### 4.1.3 META-GENERALIZATION TO DEEPER NETWORKS

In this section, we evaluate LO meta-generalization to deeper networks. Specifically, we increase the number of layers used in MLP, ViT, and LM tasks from 3 to 16, while being sure to select models that have widths within the meta-training range ($128 - 1024$) to avoid confounding the results. Figure 5 reports the performance of our multi-task learned optimizers on deeper networks. We observe that both $\mu$LO$_M$ and $\mu$VeLO$_M$ optimize stably throughout and generally outperform their counterparts, LO$_M$ and VeLO$_M$, by the end of training on each task, despite being meta-trained on MLPs of exactly the same depth. Moreover, LO$_M$ immediately diverges when optimizing the deep MLP while $\mu$LO$_M$ experience no instability. Similarly, VeLO$_M$ diverges on ViTs and Transformers,

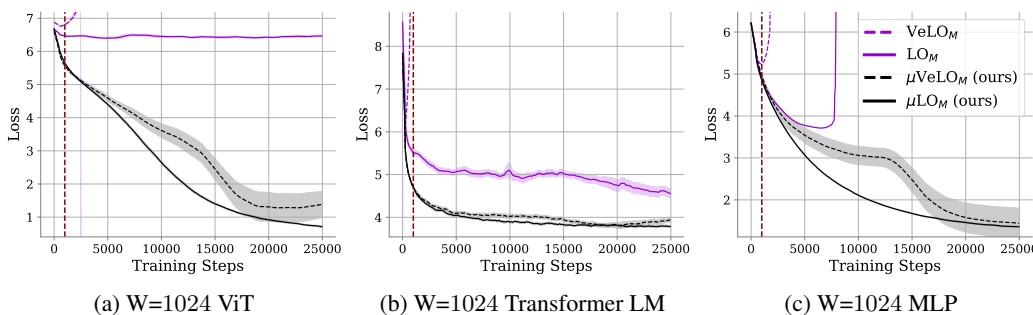

(a) W=1024 ViT      (b) W=1024 Transformer LM      (c) W=1024 MLP

Figure 6: **Evaluating generalization capabilities of $\mu$LOs to longer training horizons**. We plot average training loss over 5 seeds with standard error bars. All optimizers are meta-trained for 1000 steps of training (dotted red line), therefore, any optimization beyond 1000 steps is considered out-of-distribution. We observe that $\mu$LOs seamlessly generalize to training horizons $25\times$ longer than meta-training. In contrast, the best performing SP LO fails to decrease training loss (a), decreases it but suffers instabilities (b), or diverges after 8000 steps (c).

while $\mu$VeLO$_M$ performs well, especially on ViTs. This is remarkable as, unlike width, there is no theoretical justification for $\mu$P's benefit to deeper networks. We hypothesize that $\mu$P's stabilizing effect on the optimizee's activations leads to this improvement generalization.

*In summary, we find, empirically, that using $\mu$P during meta-training benefits the generalization of learned optimizers, including VeLO, to deeper networks.*

### 4.1.4 META-GENERALIZATION TO LONGER TRAINING HORIZONS

In this subsection, we empirically evaluate the capability of $\mu$LOs to generalize to much longer training horizons than those seen during meta-training. Specifically, we use $\mu$LO$_M$ and LO$_M$ as well as $\mu$VeLO$_M$ and VeLO$_M$ to train three networks with width $w = 1024$: a 3-layer MLP, ViT on $32 \times 32 \times 3$ ImageNet and a 3-layer Transformer for autoregressive language modeling on LM1B. Each model is trained for $25,000$ steps ($25\times$ the longest unroll seen at meta-training time). Figure 6 reports the training loss averaged over 5 random seeds. We observe that $\mu$LO$_M$ and $\mu$VeLO$_M$ stably decrease training loss over time for each task, while LO$_M$ and VeLO$_M$ fail to decrease training loss (a), decreases it but suffers instabilities (b), or diverges after 8000 steps (c). These results suggest that generalization to longer training horizons is another benefit of using $\mu$P with learned optimizers.

*In summary, we find, empirically, that using $\mu$P during meta-training significantly benefits the generalization of learned optimizers to longer training horizons.*

## 5 LIMITATIONS

While we have conducted a systematic empirical study and shown strong results within the scope of our study, there are some of limitations of our work. Specifically, (1) we do not meta-train on tasks other than MLPs for image classification and we do not provide evaluation of models wider than 8192 for MLPs and 3072/12288 (hidden/FFN size) for transformers due to computational constraints in our academic environment.

## 6 CONCLUSION

We have demonstrated that applying or $\mu$LO meta-training recipe produces optimizers with substantially improved meta-generalization properties when compared to strong baselines from previous work. Remarkably, our $\mu$LOs even surpass VeLO-4000 (meta-trained for 4000 TPU months) on wider versions of in-distribution tasks. Moreover, our experiments also show that $\mu$LOs meta-trained with our recipe generalize better to wider and deeper out-of-distribution tasks than their SP counterparts. Moreover, when evaluated on much longer training tasks, we observe that $\mu$LOs have a stabilizing effect, enabling meta-generalization to much longer unrolls ($25\times$ maximum meta-training unroll length). All of the aforementioned benefits of $\mu$LOs come at *zero* extra computational cost compared to SP LOs. Our results outline a promising path forward for low-cost meta-training of learned optimizers that can generalize to large unseen tasks.

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

## A    PROOF OF PROPOSITION 1

**Proposition 1.** *Assume that the LO $f_\phi$ is continuous around 0. Then, if $f_\phi(0) \neq 0$, the update, initialization, and pre-activation multiplier above is necessary to obtain a Maximal Update Parametrization.*

*Proof.* The update produced by $f_\phi$ is denoted $\Delta W$ and we write $\nabla W$ the corresponding gradient, so that $\Delta W = f_\phi(\nabla W)$. For the sake of simplicity, $n$ will be the output size and $d$ the feature input size of our neural network. Our goal is to satisfy the desiderata of (Yang et al., 2022, Appendix J.2). We assume our initialization follows Initialization-$\mu$ in Sec 3. Overall, our goal is to study strategy so that if $x_i = \Theta(1)$, then one needs to renormalize/initialize so that $(Wx)_i = \Theta(1)$ while $((W + \Delta W)x)_i = \Theta(1)$ so that the update is as large as possible. Note that given the assumptions on $f$, if $x = \Theta(\frac{1}{n})$, then $f(x) = \Theta(1)$.

**Output weights.**    Here, if input $x$ has some $\Theta(1)$ coordinates, we initialize $W = (w_i)_{i \leq n}$ with weights of variance 1 (which is necessary) and rescale the preactivations with $\frac{1}{n}$. For the update, we thus have that $\nabla W$ scales (coordinate wise) as $\Theta(\frac{1}{n})$ and we do not rescale the LR, given that $f_\phi(\nabla W)$ will also have coordinates in $\Theta(1)$.

**Hidden weights.**    Now, for the update, we observe that the gradient $\nabla W$ has some coordinates which scale as $\Theta(\frac{1}{n})$, due to the output renormalization choice. Thus, the LO $f_\phi(\nabla W)$ satisfies that $f(\nabla W) = \Theta(1)$, given that $f_\phi$ is continuous in 0 and satisfies $f_\phi(0) \neq 0$. Thus for the update, we need to use $\Delta W = \frac{1}{n} f_\phi(\nabla W)$ so that $\Delta W x$ is coordinate wise bounded.

**Input weights.**    In this case, the gradient has coordinates which already scale in $\Theta(\frac{1}{n})$ (due to the output renormalization) and there is no need to rescale the LR. □

## B    HAND DESIGNED OPTIMIZER HYPERPARAMETER TUNING

### B.1    TUNING $\mu$ADAM

We tune the $\mu$Adam baseline on the largest meta-training seen by our learned optimizers. $\mu$Adam$_M$ was, therefore, tuned using a 1024 width MLP for $32 \times 32 \times 3$ ImageNet classification. As mentioned in section 4, we tune the learning rate and three multipliers: input multiplier, output multiplier, and the hidden learning rate multiplier. These multipliers correspond to adding a tunable constant to the pre-activation multiplier for input weights, the pre-activation multiplier for output weights, and the Adam LR for hidden weights (e.g., in Table 8 of Yang et al. (2022)). Specifically, we search for the learning rate in $\{0.1, 0.01, 0.001, 0.0001\}$ and for each multiplier in $\{2^{-4}, 2^{-2}, 1, 2^2, 2^4\}$. This results in a grid search of 500 configurations, whose optimal values are reported in table 3.

Table 3: **Best hyperparameters values for $\mu$Adam baseline**. $\mu$Adam is tuned to optimize 3-layer W= 1024 MLP for $32 \times 32 \times 3$ ImageNet classification, while $\mu$Adam (re-tuned) is tuned on 3-layer W= 384 ViT for $32 \times 32 \times 3$ ImageNet classification.

| Optimizer | LR | Input Multiplier | Output Multiplier | Hidden LR Multiplier |
|---|---|---|---|---|
| $\mu$Adam | 0.1 | 0.25 | 0.25 | 4 |
| $\mu$Adam (re-tuned) | 0.000702 | 0.9 | 0.95 | 0.01 |

### B.2    TUNING ADAMW

We tune the AdamW baseline on the largest meta-training seen by our learned optimizers. AdamW was, therefore, tuned using a 1024 width MLP for $32 \times 32 \times 3$ ImageNet classification. As mentioned in section 4, we tune the learning rate, betas, and weight decay: LR, $\beta_1$, $\beta_2$, and the weights decay. Specifically, we search over the values of each hyperparameter reported in Table 4. This results in a grid search of 500 configurations, whose optimal values are reported in table 5.

Table 4: **Grid search values used for AdamW**. Similar to the $\mu$Adam baseline, we tune all optimizers on a 3-layer W= 1024 MLP ImageNet classification task and use a budget of approximately 500 total runs. We tune LR, $\beta_1$, $\beta_2$, and weight decay to minimize training loss after 1000 steps.

| Optimizer | LR | $\beta_1$ | $\beta_2$ | weight decay | Total runs |
|---|---|---|---|---|---|
| SP AdamW | Log Sample 14 from $[10^{-5}, 0.1]$ | {0.9,0.95,0.99} | {0.95,0.99,0.999} | {0.1,0.01,0.001,0.0001} | 504 |

Table 5: **Optimal Hyperparameters Found AdamW**. Similar to the $\mu$Adam baseline, we tune all optimizers on a 3-layer W= 1024 MLP ImageNet classification task and use a budget of approximately 500 total runs.

| Optimizer | LR | $\beta_1$ | $\beta_2$ | weight decay | Total runs |
|---|---|---|---|---|---|
| SP AdamW | 0.000702 | 0.9 | 0.95 | 0.0001 | 504 |

## C   META-TRAINING WITH $\mu$LOS

**General meta-training setup for small_fc_lopt**   Each small_fc_lopt (Metz et al., 2022a) learned optimizer is meta-trained for 5000 steps of gradient descent using AdamW (Loshchilov and Hutter, 2019) and a linear warmup and cosine annealing schedule. We using PES (Vicol et al., 2021) to estimate meta-gradients with a truncation length of 50 steps and sampling 8 perturbations per task at each step with standard deviation 0.01. For the inner optimization task, we used a maximum unroll length of 1000 iterations; that is, all our learned optimizers see at most 1000 steps of the inner optimization problem during meta-training. Unlike with $\mu$Adam, we do not tune the $\mu$P multipliers when meta-training $\mu$LO$_S$ and $\mu$LO$_M$, instead, we set the all to 1. All optimizers are meta-trained on a single A6000 GPU. $\mu$LO$_S$ and LO$_S$ take 8 hours each to meta-train, while $\mu$LO$_M$ and LO$_M$ take 103 hours.

**General meta-training setup for VeLO**   Each VeLO (Metz et al., 2022a) learned optimizer is meta-trained for 45000 steps of gradient descent using AdamW (Loshchilov and Hutter, 2019) and a linear warmup and cosine annealing schedule. We using PES (Vicol et al., 2021) to estimate meta-gradients with a truncation length of 20 steps and sampling 8 perturbations per task at each step with standard deviation 0.01. For the inner optimization task, we used a maximum unroll length of 1000 iterations; that is, all our learned optimizers see at most 1000 steps of the inner optimization problem during meta-training. Unlike with $\mu$Adam, we do not tune the $\mu$P multipliers when meta-training $\mu$LO$_S$ and $\mu$LO$_M$, instead, we set the all to 1. All optimizers are meta-trained on a single A6000 GPU. $\mu$VeLO$_M$ and VeLO$_M$ take 250 hours to meta-train.

**Meta-training hyperparameters for small_fc_lopt in $\mu$P**   While there are very few differences between $\mu$LOs and SP LOs, the effective step size for hidden layers is changed (see eq. 3) which could alter the optimal meta-training hyperparameters. Consequently, we conduct an ablation study on hyper-parameters choices for $\mu$LO$_S$. Specifically, using AdamW and gradient clipping with a linear warmup and cosine annealing LR schedule, we meta-train $\mu$LO$_S$ to train 3-layer width 128 MLPs to classify $32 \times 32 \times 3$ ImageNet Images. By default, we warmup linearly for 100 steps to a maximum learning rate of $3e - 3$ and anneal the learning rate for 4900 steps to a value of $1e - 3$ with $\lambda_1 = 0.001$ (from equation 3) and sampling 8 perturbations per step in PESVicol et al. (2021). The above ablation varies the maximum learning rate $\in \{1e - 2, 3e - 3, 1e - 3\}$ (always using 100 steps of warmup and decaying to $0.3 \times$MaxLR), $\lambda_1 \in \{0.001, 0.01, 0.1\}$, the number of steps (5k or 10k), and the number of perturbations (8 or 16). We observe that using all default values except for $\lambda_1 = 0.01$ yields one of the best solutions while being fast to train and stable during meta-training. We, therefore, select these hyperparameters to meta-train $\mu$LO$_S$ and $\mu$LO$_M$.

**Meta-training hyperparameters for VeLO in $\mu$P**   Unlike small_fc_lopt, we do not find it necessary to $\lambda_1$ from its default value. However, we do remove the multiplication by the current parameter norm used in the update equation to VeLO as it causes meta-training problems when initializing tensors to zero.

$\mu$**P at Meta-training time** While we use the same $\mu$P at meta-training and testing time, it is important to consider meta-training tasks that have similar training trajectories to their infinite width counterparts. In (Yang et al., 2022), authors provide discussions of these points for zero-shot hyperparameter transfer. Two notable guidelines are to initialize the output weight matrix to zero (as it will approach zero in the limit) and to use a relatively large key size when meta-training transformers. For all our tasks, we initialize the network's final layer to zeros. While we do not meta-train on transformers, we suspect that the aforementioned transformer-specific guidelines may be useful.

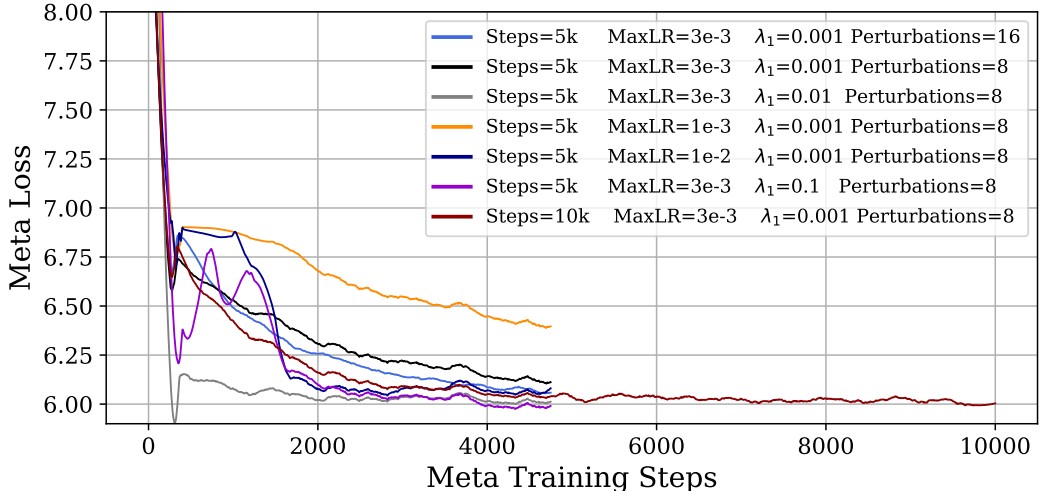

Figure 7: **Ablating Meta-training Hyperparameter for $\mu$LO$_S$.** All curves show a single meta-training run. Using AdamW with a linear warmup and cosine annealing schedule, we meta-train $\mu$LO$_S$ to train 3-layer width 128 MLPs for classifying $32 \times 32 \times 3$ ImageNet Images. By default, we warmup linearly for 100 steps to a maximum learning rate of $3e - 3$ and anneal the learning rate for $4900$ steps to a value of $1e - 3$ with $\lambda_1 = 0.001$ (from equation 3) and sampling 8 perturbations per step in PESVicol et al. (2021). The above ablation varies the maximum learning rate $\in \{1e-2, 3e-3, 1e-3\}$ (always using 100 steps of warmup and decaying to $0.3 \times$MaxLR), $\lambda_1 \in \{0.001, 0.01, 0.1\}$, the number of steps (5k or 10k), and the number of perturbations (8 or 16). We observe that using all default values except for $\lambda_1 = 0.01$ yields one of the best solutions while being fast to train and stable during meta-training.

## D FEATURES OF THE LEARNED OPTIMIZER

Table 6: **Per-parameter features used as input to our learned optimizers.** All the coefficients, $\beta_i$, are learnable parameters adjusted during meta-optimization. We replicate the table of (Joseph et al., 2023) for convenience.

| Description | value |
|---|---|
| parameter value | $w_t$ |
| 3 momentum values with coefficients $\beta_1, \beta_2, \beta_3$ | $m_{t,i} = \beta_i m_{t-1,i} + (1 - \beta_i)g_t$ |
| second moment value computed from $g_t$ with decay $\beta_4$ | $v_t = \beta_4 v_{t-1} + (1 - \beta_4)g_t^2$ |
| 3 values consisting of the three momentum values normalized by the square root of the second moment | $\frac{m_{t,i}}{\sqrt{v}}$ |
| the reciprocal square root of the second moment value | $\frac{1}{\sqrt{v}}$ |
| 3 $\Delta_t$ Adafactor normalized values | $g_t \times$ ROW FACTOR $\times$ COLUMN FACTOR |
| 3 tiled Adafactor row features with coefficients $\beta_5, \beta_6, \beta_7$, computed from $g_t$ | $r_{t,i} = \beta_i r_{t-1,i} + (1 - \beta_i)$ROW_MEAN$(\Delta_t^2)$ |
| 3 tiled Adafactor column feature with coefficients $\beta_5, \beta_6, \beta_7$ computed from $g_t$ | $c_{t,i} = \beta_i c_{t-1,i} + (1 - \beta_i)$COL_MEAN$(\Delta_t^2)$ |
| the reciprocal square root of the previous 6 features | $\frac{1}{\sqrt{r_{t,i} \text{ OR } c_{t,i}}}$ |
| 3 $m$ Adafactor normalized values | $m_{t,i} \times$ ROW FACTOR $\times$ COLUMN FACTOR |

# E    LIST OF META-TESTING TASKS

Table 7 reports the configuration of different testing tasks used to evaluate our optimizers. We note that we do not augment the ImageNet datasets we use in any way except for normalizing the images. We tokenize LM1B using a sentence piece tokenizer(Kudo and Richardson, 2018) with 32k vocabulary size. All evaluation tasks are run on A6000 $48$BG or A100 $80$GB GPUs for $5$ random seeds.

Table 7: **Meta-testing settings.** We report the optimization tasks we will use to evaluate the LOs of Table 1.

| Identifier | Dataset | Model | Depth | Width | Attn. Heads | FFN Size | Batch Size | Sequence Length |
|---|---|---|---|---|---|---|---|---|
| IN32$\mathcal{T}^{\mathrm{MLP}}_{(3,128)}$ | $32 \times 32 \times 3$ ImageNet | MLP | 3 | 128 | – | – | 4096 | – |
| IN32$\mathcal{T}^{\mathrm{MLP}}_{(3,256)}$ | $32 \times 32 \times 3$ ImageNet | MLP | 3 | 256 | – | – | 4096 | – |
| IN32$\mathcal{T}^{\mathrm{MLP}}_{(3,512)}$ | $32 \times 32 \times 3$ ImageNet | MLP | 3 | 512 | – | – | 4096 | – |
| IN32$\mathcal{T}^{\mathrm{MLP}}_{(3,1024)}$ | $32 \times 32 \times 3$ ImageNet | MLP | 3 | 1024 | – | – | 4096 | – |
| IN32$\mathcal{T}^{\mathrm{MLP}}_{(3,2048)}$ | $32 \times 32 \times 3$ ImageNet | MLP | 3 | 2048 | – | – | 4096 | – |
| IN32$\mathcal{T}^{\mathrm{MLP}}_{(3,4096)}$ | $32 \times 32 \times 3$ ImageNet | MLP | 3 | 4096 | – | – | 4096 | – |
| IN32$\mathcal{T}^{\mathrm{MLP}}_{(3,8192)}$ | $32 \times 32 \times 3$ ImageNet | MLP | 3 | 8192 | – | – | 4096 | – |
| IN64$\mathcal{T}^{\mathrm{MLP}}_{(3,128)}$ | $64 \times 64 \times 3$ ImageNet | MLP | 3 | 128 | – | – | 4096 | – |
| IN64$\mathcal{T}^{\mathrm{MLP}}_{(3,256)}$ | $64 \times 64 \times 3$ ImageNet | MLP | 3 | 256 | – | – | 4096 | – |
| IN64$\mathcal{T}^{\mathrm{MLP}}_{(3,512)}$ | $64 \times 64 \times 3$ ImageNet | MLP | 3 | 512 | – | – | 4096 | – |
| IN64$\mathcal{T}^{\mathrm{MLP}}_{(3,1024)}$ | $64 \times 64 \times 3$ ImageNet | MLP | 3 | 1024 | – | – | 4096 | – |
| IN64$\mathcal{T}^{\mathrm{MLP}}_{(3,2048)}$ | $64 \times 64 \times 3$ ImageNet | MLP | 3 | 2048 | – | – | 4096 | – |
| IN64$\mathcal{T}^{\mathrm{MLP}}_{(3,4096)}$ | $64 \times 64 \times 3$ ImageNet | MLP | 3 | 4096 | – | – | 4096 | – |
| C10$\mathcal{T}^{\mathrm{MLP}}_{(3,128)}$ | $32 \times 32 \times 3$ Cifar-10 | MLP | 3 | 128 | – | – | 4096 | – |
| C10$\mathcal{T}^{\mathrm{MLP}}_{(3,256)}$ | $32 \times 32 \times 3$ Cifar-10 | MLP | 3 | 256 | – | – | 4096 | – |
| C10$\mathcal{T}^{\mathrm{MLP}}_{(3,512)}$ | $32 \times 32 \times 3$ Cifar-10 | MLP | 3 | 512 | – | – | 4096 | – |
| C10$\mathcal{T}^{\mathrm{MLP}}_{(3,1024)}$ | $32 \times 32 \times 3$ Cifar-10 | MLP | 3 | 1024 | – | – | 4096 | – |
| C10$\mathcal{T}^{\mathrm{MLP}}_{(3,2048)}$ | $32 \times 32 \times 3$ Cifar-10 | MLP | 3 | 2048 | – | – | 4096 | – |
| C10$\mathcal{T}^{\mathrm{MLP}}_{(3,4096)}$ | $32 \times 32 \times 3$ Cifar-10 | MLP | 3 | 4096 | – | – | 4096 | – |
| C10$\mathcal{T}^{\mathrm{MLP}}_{(3,8192)}$ | $32 \times 32 \times 3$ Cifar-10 | MLP | 3 | 8192 | – | – | 4096 | – |
| $\mathcal{T}^{\mathrm{ViT}}_{(3,192)}$ | $32 \times 32 \times 3$ ImageNet | ViT | 3 | 192 | 3 | 768 | 4096 | – |
| $\mathcal{T}^{\mathrm{ViT}}_{(3,384)}$ | $32 \times 32 \times 3$ ImageNet | ViT | 3 | 384 | 6 | 1536 | 4096 | – |
| $\mathcal{T}^{\mathrm{ViT}}_{(3,768)}$ | $32 \times 32 \times 3$ ImageNet | ViT | 3 | 768 | 8 | 3072 | 4096 | – |
| $\mathcal{T}^{\mathrm{ViT}}_{(3,1024)}$ | $32 \times 32 \times 3$ ImageNet | ViT | 3 | 1024 | 8 | 4096 | 4096 | – |
| $\mathcal{T}^{\mathrm{ViT}}_{(3,2048)}$ | $32 \times 32 \times 3$ ImageNet | ViT | 3 | 2048 | 16 | 8192 | 4096 | – |
| $\mathcal{T}^{\mathrm{ViT}}_{(3,3072)}$ | $32 \times 32 \times 3$ ImageNet | ViT | 3 | 3072 | 16 | 12288 | 4096 | – |
| $\mathcal{T}^{\mathrm{LM}}_{(3,192)}$ | LM1B, $32k$ Vocab | Transformer LM | 3 | 192 | 3 | 768 | 4096 | 64 |
| $\mathcal{T}^{\mathrm{LM}}_{(3,384)}$ | LM1B, $32k$ Vocab | Transformer LM | 3 | 384 | 6 | 1536 | 4096 | 64 |
| $\mathcal{T}^{\mathrm{LM}}_{(3,768)}$ | LM1B, $32k$ Vocab | Transformer LM | 3 | 768 | 8 | 3072 | 4096 | 64 |
| $\mathcal{T}^{\mathrm{LM}}_{(3,1024)}$ | LM1B, $32k$ Vocab | Transformer LM | 3 | 1024 | 8 | 4096 | 4096 | 64 |
| $\mathcal{T}^{\mathrm{LM}}_{(3,2048)}$ | LM1B, $32k$ Vocab | Transformer LM | 3 | 2048 | 16 | 8192 | 4096 | 64 |
| $\mathcal{T}^{\mathrm{LM}}_{(3,3072)}$ | LM1B, $32k$ Vocab | Transformer LM | 3 | 3072 | 16 | 12288 | 4096 | 64 |
| $\mathcal{DT}^{\mathrm{MLP}}_{(16,512)}$ | $32 \times 32$ ImageNet | MLP | 16 | 512 | – | – | 4096 | – |
| $\mathcal{DT}^{\mathrm{ViT}}_{(16,192)}$ | $32 \times 32$ ImageNet | ViT | 16 | 192 | 3 | 768 | 4096 | – |
| $\mathcal{DT}^{\mathrm{LM}}_{(16,192)}$ | LM1B | Transformer LM | 16 | 192 | 3 | 768 | 4096 | – |

## F    TASK-SPECIFIC TUNED $\mu$ADAM

In this section, we evaluate the meta-generalization performance of $\mu\text{LO}_M$ and $\mu\text{VeLO}_M$ relative to $\mu$Adam and $\mu$Adam (re-tuned) on a w=3072 ViT $32 \times 32$ ImageNet task. $\mu$Adam is tuned on a width=1024 MLP task for 500 trials and $\mu$Adam (re-tuned) is tuned on a width=384 ViT task for 500 trials. The hyperparameters of these baselines are reported in table 3. In figure 8, we observe that $\mu$Adam is outperformed by $\mu$Adam (re-tuned) as expected. We note that $\mu$Adam (re-tuned) is tuned in the $\mu-$transfer setting of Yang et al. (2022) where one tunes on a smaller width version of the target task. This experiment allows us to assess whether $\mu$LO out-of-distribution can outperform $\mu$-transfer in-distribution. Despite being evaluated out-of-distribution, $\mu\text{LO}_M$ and $\mu\text{VeLO}_M$ outperformed the re-tuned $\mu$Adam baseline on the width 3072 ViT task. These results demonstrate that the $\mu$LO framework has the potential to show strong transfer even for unseen tasks.

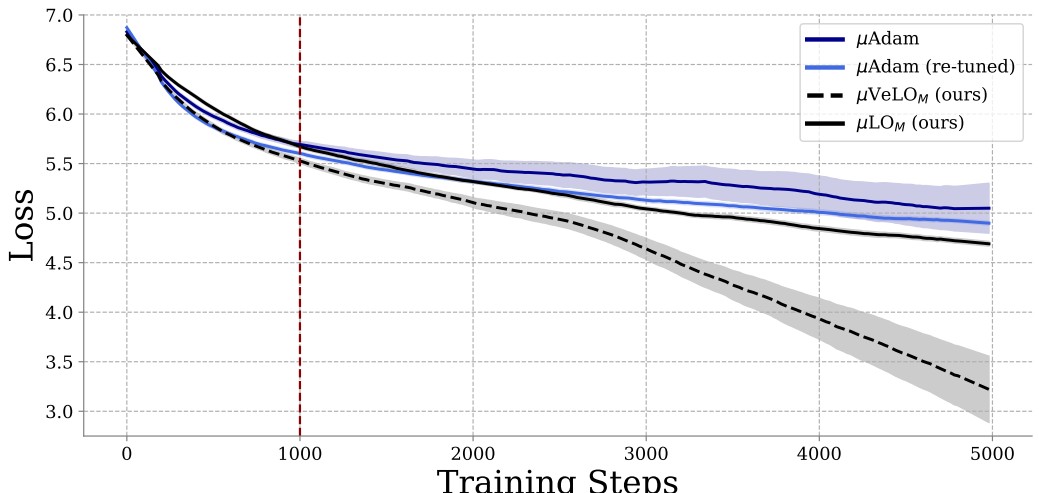

Figure 8: **Comparing the performance of $\mu$LOs to $\mu$Adam on a width** 3072 **ViT task.** Each curve reports the mean training loss over 5 trials. Error bars report standard error. $\mu$Adam was tuned on a width 1024 MLP task for 500 trials, while $\mu$Adam (re-tuned) was tuned on a width 384 ViT task for 500 trials. We observe that the re-tuned $\mu$Adam baseline bests its counterpart, but is outperformed by our $\mu$LOs.

## G  RESULTS FOR RESNETS AND PLAIN RESNETS

Prior work has demonstrating the difficulty of optimizing deep networks without residual connections Li et al. (2018); He et al. (2016). Specifically, Li et al. (2018) demonstrates that the loss landscape is much smoother for ResNets than plain ResNets. Such pernicious loss landscapes could pose problems for gradient-based optimizers. Could this be the case for learned optimizers? How do $\mu$LOs affect this? In this section, we answer this question by ablating the performance of $\mu$LOs and SP learned optimizers on plain and residual networks. Figures 9 reports the training curves for ResNets (subfigure a) and plain ResNets (subfigure b). We observe that VeLO$_M$ immediately diverges in both cases, LO$_M$ initially decreases the loss faster than $\mu$LO$_M$ and $\mu$VeLO$_M$, but it eventually stagnates and is surpassed by bot $\mu$LOs, and $\mu$LOs monotonically decrease the loss during the first 5000 steps of training.

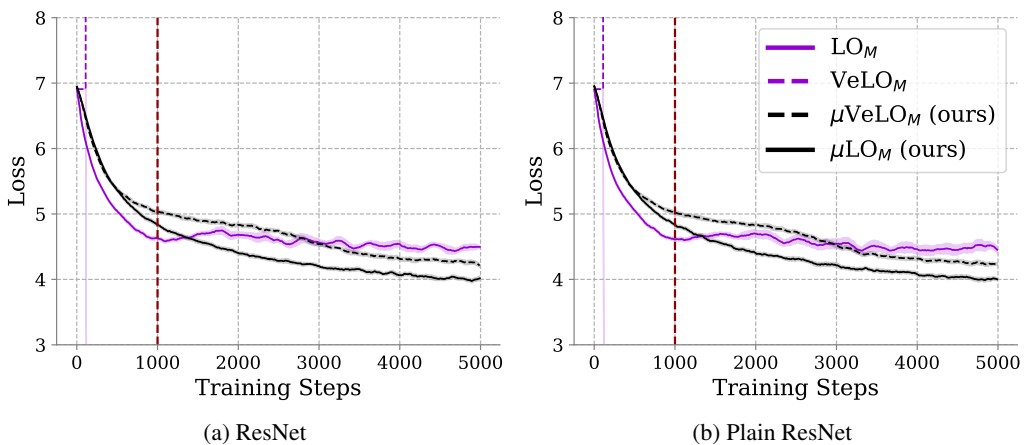

(a) ResNet                                    (b) Plain ResNet

Figure 9: **Performance of Deep Plain and Residual Networks.** We report the training loss for a depth 24 and width 256 plain and residual networks. We observe similar trends for both residual and plan networks: 1) VeLO$_M$ immediately diverges in both cases, 2) LO$_M$ initially decreases the loss faster than $\mu$LO$_M$ and $\mu$VeLO$_M$, but it eventually stagnates and is surpassed by bot $\mu$LOs, and 3) $\mu$LOs monotonically decrease the loss during the first 5000 steps of training. Each curve is an average over 5 trials. The shaded regions denote standard error.

# H    EXTENDED GENERALIZATION TO LONGER UNROLLS FOR $\mu$VELO

In this section, we extend our meta-generalization results for longer unrolls. Specifically, we verify whether $\mu$VeLO can generalize beyond $25\times$ the meta-training unroll length for a width $1024$ ViT to ImageNet task as its training curve in figure 6 (a) seems to slightly increase toward the end of training. It is important to note that the VeLO architecture takes as input the number of training steps remaining, thus, requiring the user to specify the total number of training steps (`total_steps`) a-priori (e.g. as is done for many LR schedules in practice). Therefore, at each step, VeLO's LSTM is conditioned on an embedding that provides the number of training steps remaining, allowing it to learn a schedule. Previous work analyzing VeLO-4000's behavior has noted that changing the value of the `total_steps` hyperparameter leads to variable performance Rezk et al. (2023). Specifically, they found that increasing the value of `total_steps` does not always lead to better performance Rezk et al. (2023). Figure 10 demonstrates that $\mu$VeLO$_M$ can successfully optimize a width $1024$ ViT to classify ImageNet images (same task as Figure 6 (a)) for $40,000$ training steps. However, we note that it underperforms $\mu$VeLO$_M$ using `total_steps` $= 25,000$. This is similar to what was found in previous work for VeLO-4000 Rezk et al. (2023).

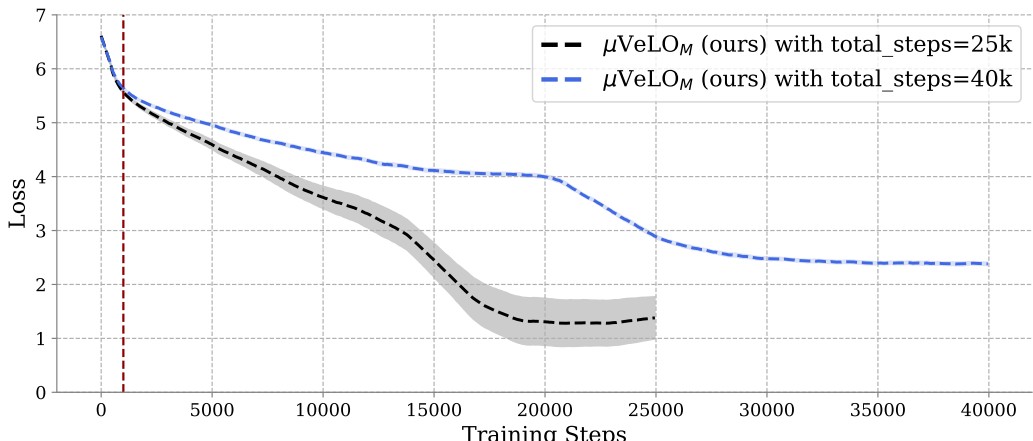

Figure 10: **Comparing the performance of $\mu$VeLO$_M$ on a width $1024$ ViT ImageNet task when the total training steps are set to 25,000 and 40,000.** Each curve reports the mean training loss over 5 trials. Error bars report standard error. We observe that both decrease the loss throughout training, except after iteration 20,000 for $\mu$VeLO$_M$ with `total_steps` $= 25k$, which seems to suffer from a very slight increase in loss. Notably, similar to what is shown in previous work Rezk et al. (2023) for VeLO-4000, $\mu$VeLO$_M$ using `total_steps` $= 40k$ underperforms $\mu$VeLO$_M$ using `total_steps` $= 25k$.

# I   COORDINATE EVOLUTION OF MLP LAYERS IN $\mu$P FOR DIFFERENT OPTIMIZERS

The following section presents the continuation of our experiments comparing pre-activation growth during training for SP LOs and $\mu$LOs with different meta-trainnig recipes, SP adam, and $\mu$Adam.

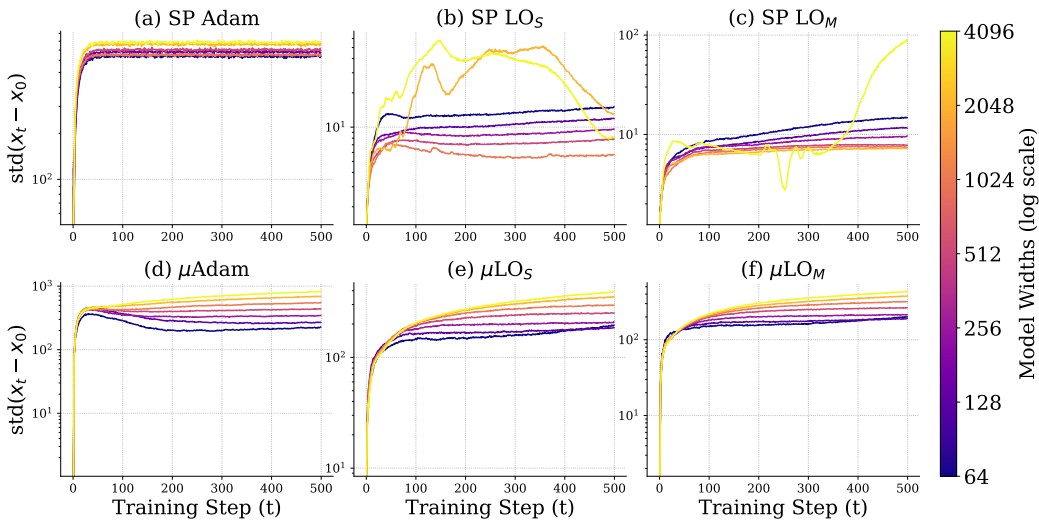

Figure 11: **Layer 0 pre-activations behave harmoniously in $\mu$P for LOs and Adam alike.** We report the evolution of coordinate-wise standard deviation between the difference of initial and current second-layer pre-activations. We observe that all models parameterized in $\mu$P enjoy stable coordinates across widths, while the pre-activations of larger-width models in SP blow up after a number of training steps. All plots report these metrics for the first 500 steps of a single training run.

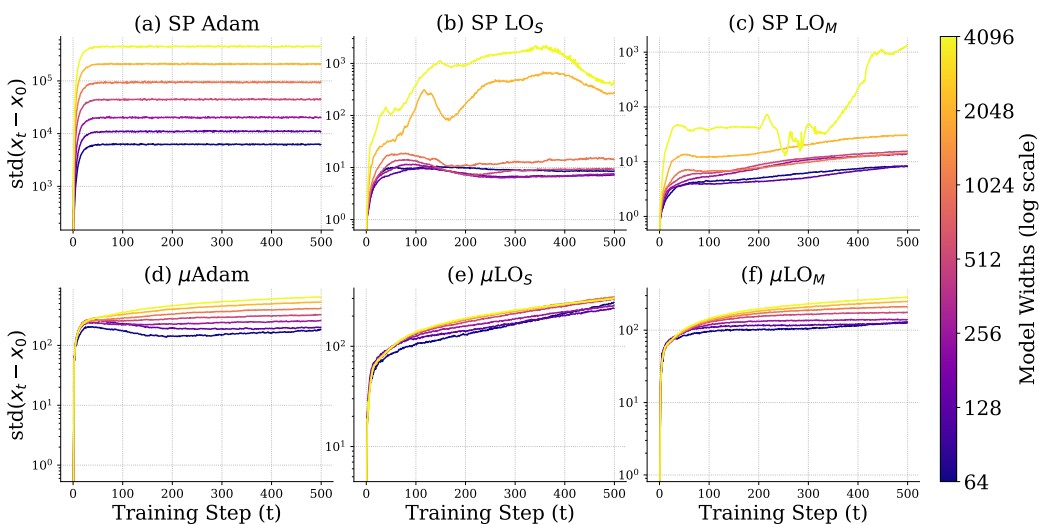

Figure 12: **Layer 1 pre-activations behave harmoniously in $\mu$P for LOs and Adam alike.** We report the evolution of coordinate-wise standard deviation between the difference of initial and current second-layer pre-activations. We observe that all models parameterized in $\mu$P enjoy stable coordinates across widths, while the pre-activations of larger-width models in SP blow up after a number of training steps. All plots report these metrics for the first 500 steps of a single training run.

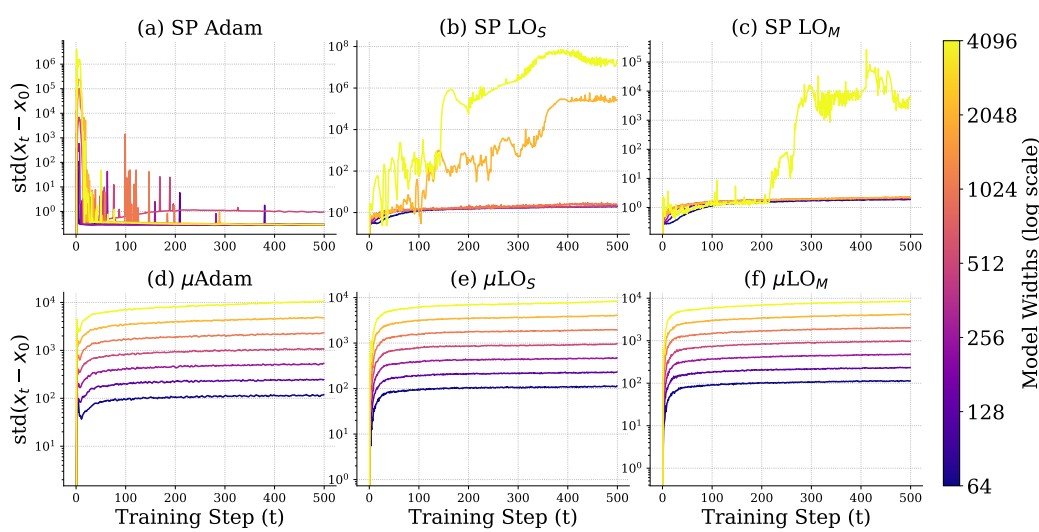

Figure 13: **Layer 3 pre-activations behave harmoniously in $\mu$P for LOs and Adam alike.** We report the evolution of coordinate-wise standard deviation between the difference of initial and current second-layer pre-activations. We observe that all models parameterized in $\mu$P enjoy stable coordinates across widths, while the pre-activations of larger-width models in SP blow up after a number of training steps. All plots report these metrics for the first 500 steps of a single training run.

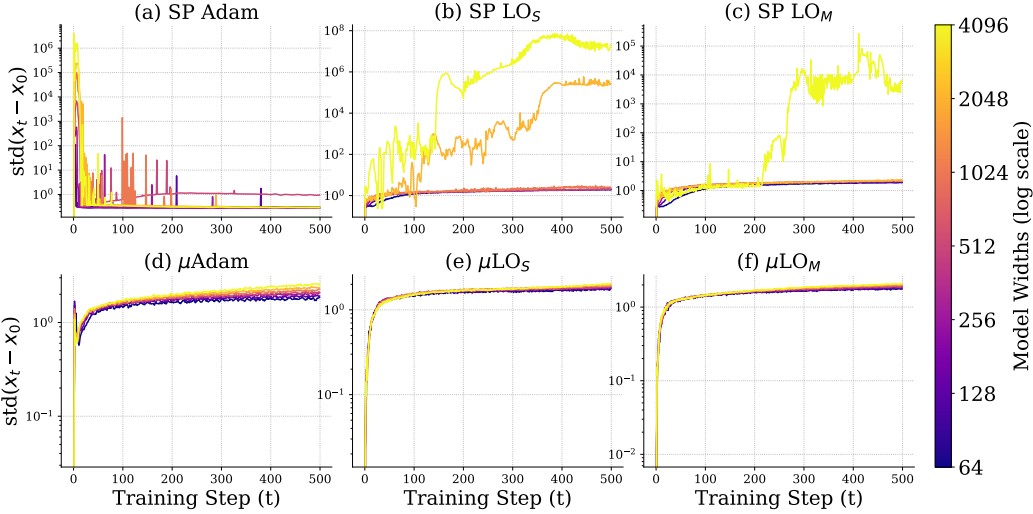

Figure 14: **Logits behave harmoniously in $\mu$P for LOs and Adam alike.** We report the evolution of coordinate-wise standard deviation between the difference of initial and current second-layer pre-activations. We observe that all models parameterized in $\mu$P enjoy stable logits across widths, while the pre-activations of larger-width models in SP blow up after a number of training steps. All plots report these metrics for the first 500 steps of a single training run.

