# OpenReview forum: "$\mu$LO: Compute-Efficient Meta-Generalization of Learned Optimizers"
_ICLR.cc/2025/Conference — Submitted to ICLR 2025_

### Official Review · Reviewer_oHHy · 2024-10-19

**Soundness:** 3
**Presentation:** 3
**Contribution:** 2
**Rating:** 6
**Confidence:** 4

**Summary:**

In this paper, the authors aim to improve on the challenging task of improving the meta-generalization of Learned Optimizers (LOs) to longer optimization trajectories and larger base models while maintaining compute efficiency. The authors derive Maximal Update Parametrization for two popular learned optimizer architectures, enabling zero-shot hyperparameter transfer from smaller models to larger ones. A simple training recipe for µ-parameterized LO is proposed which leads to substantial improvement in meta-generalization to wider networks, deeper networks, and longer training horizons with orders of magnitude less computational cost.

**Strengths:**

• Overall, I think the paper is well-written and addresses a very meaningful problem in the meta-learned optimizer.

• The empirical performance of the proposed algorithm is very good (Figures 4, 5, and 6), and it is particularly impressive that they were able to get such good results using so little compute relative to VeLO (which is extremely expensive …).

**Weaknesses:**

• The proposed method is quite simple, essentially, describing a way to construct a model such that the meta-learned optimizer can generalize to larger-width networks, deeper networks, and longer training trajectories. As far as I am aware the novelty of this paper is relatively limited since most of the findings are originally derived from [1] where Maximal Update Parametrizations was originally proposed. Therefore, from my perception, this appears to be a straightforward application of that work. Could the authors please elaborate on the technical novelty, of this paper? If the technical novelty is more than just an application of Maximal Update Parametrizations, I am happy to increase my score.

• The paper could be made more self-contained. I had to read [1] before really understanding much of the terminology and language being used in this paper.

[1] Yang, Greg, et al. "Tensor programs v: Tuning large neural networks via zero-shot hyperparameter transfer." NeurIPS2021

**Questions:**

• From my understanding [1] shows how to initialize the network such that you can scale the width of the network while maintaining optimal hyper-parameters. This paper seems to suggest that using this strategy also allows you to scale the depth of the network and the optimization trajectory. Why is that the case? Can the authors please add some discussion about that and clarify more theoretically why the results are the way they are.

---

> ### Author Response · Authors · 2024-11-22
> **General comments and replies to requested changes**
>
> Thank you for taking the time to review our paper. We are pleased to hear oHHy believes our paper is well written, that oHHy thinks it addresses an important problem for learned optimization, and that oHHy finds the empirical performance of our approach to be impressive.
>
> We now address the reviewer's concerns. Each bolded paragraph title refers to one or multiple concerns that have been grouped together. All line numbers refer to the originally submitted manuscript (not the updated version).
>
>
> **Technical contribution beyond applying $\mu$P** We agree that the change of meta-training LOs in $\mu$P is simple, however, the simplicity of our approach is one of its greatest strengths. This small but non-trivial change relative to the existing meta-training framework in learned optimization enables substantial improvements to meta-generalization with no added computational cost to the learned optimizer and no restriction on the problems that can be considered (e.g., any optimizee can be considered in the $\mu$LO framework). As such, it represents a net improvement over the existing meta-training framework in learned optimization [4.1,4.2,4.3].
>
> Another contribution of our work is alleviating the limitations of $\mu$-transfer [4.4], itself. As outlined in lines 226-245, $\mu$-transfer increases the complexity of hyperparameter optimization (HPO), while requiring re-tuning for each new task and dataset considered—without the possibility of transferring HPO knowledge from one task to another (e.g. $\mu$Adam underperforms in heldout tasks Fig 4. (c),(d),(e)). In contrast, $\mu$LOs allows amortizing the cost of future HPO during meta-training at low cost (in contrast to [4.1], which also amortizes HPO) and can benefit from knowledge sharing across multiple tasks during meta-training, unlike $\mu$-transfer.
>
> Therefore, learned optimization helps further alleviate the tuning costs of $\mu$-transfer for new problems, and parameterizing optimizees in $\mu$P improves the generalization of LOs from small to large tasks. As such, $\mu$ learned optimization is neither a simple application of learned optimization to $\mu$-transfer, nor a simple application of $\mu$-transfer to LOs. It constitute a new meta-training framework for learned optimization that is of interest for amortizing HPO and improving the meta-generalization of LOs.
>
>
> **The paper could be more self-contained with respect to [4.4]** We have updated the manuscript by adding a paragraph to section 3.2 that adds more background about $\mu$P to the text. If oHHy could provide more details on anything they believe is still missing we would be grateful.
>
> **Question regarding generalization to depth and longer unrolls** Our observations of stronger generalization to longer unrolls and deeper networks are purely empirical.  We have amended the manuscript to emphasize that these findings are only empirical. [4.4] also shows similar empirical findings for longer and deeper versions of the tasks they tune on. Note that our findings for $\mu$LOs extend beyond the meta-training tasks to deeper and longer out-of-distribution tasks (e.g. ViT and LM in Figures 5 and 6).
>
>
> ---
> **Local References**
>
>
> [4.1] [VeLO: Training Versatile Learned Optimizers by Scaling Up, Metz et al.]
>
> [4.2] [Tasks, stability, architecture, and compute: Training more effective learned optimizers, and using them to train themselves, Metz et al.]
>
> [4.3] [Practical Tradeoffs Between Memory, Compute, and Performance in Learned Optimizers, Metz et al., CoLLAs 2022]
>
> [4.4] [Yang et al., Tensor Programs V: Tuning Large Neural Networks via Zero-Shot Hyperparameter Transfer, Neurips 2021]

---

> > ### Comment · Reviewer_oHHy · 2024-11-27
> >
> > I have read through all the reviewer and author comments, and although I still have some lingering thoughts about the novelty, I think the author/s did a good job improving the paper during the discussion phase. In general, I think this paper would be of use to the community, so I have raised my score, and I suggest acceptance for this paper.

---

### Official Review · Reviewer_umKW · 2024-11-01

**Soundness:** 3
**Presentation:** 3
**Contribution:** 3
**Rating:** 6
**Confidence:** 3

**Summary:**

This paper proposed new optimizers to overcome the generalizability issue of the existing hand-crafted learned optimizers. It showed extensive experimental results so that the proposed optimizers improve meta-generalization to wider unseen tasks in most cases.

**Strengths:**

*	Despite some cases where the proposed optimizers do not show the best performance, the “overall” results show that the proposed optimizers show better performance and better stability.


*	It is interesting and informative that it shows a stable performance in the longer horizon.

**Weaknesses:**

•	Why don’t you show results on ResNet and Plain ResNet (i.e., without residual connections)? Earlier, it has been shown that the reason why ResNet (with residual connections) works well by loss landscape – ResNet shows a smoother landscape while Plain ResNet shows a bumpy loss landscape in [1]. If the proposed optimizers can show stable performance even in Plain ResNets, I believe this work will be shown to be very effective.

•	In a nutshell, the technical addition of this work on top of the standard LP is adding 1/FAN_IN coefficient on the right term (the updated amount for the weight) in (2) if the weight is part of a hidden layer. In that sense, it is unclear how much technical contribution this work would contain. I wonder if the authors’ claim is that it shows a great advancement by adding such a small change.

•	The result in Fig 4 is confusing with regard to the goal of the work. The goal of this paper is to develop a better-generalizable optimizer. However, it did not show the best performance on the most “out-of-distribution” tasks (transformer language modeling and ViT image classification, Fig. 4.) (Although the authors argued that their LOs are meta-trained with only 0.004% of VeLO-4000.) Hence, the contribution of this work is diluted. The proposed optimizers do not show the best performance for Transformer LM, in Fig 5 either.

•	The legend in Fig 4 needs to be fixed. Either of VeLOM or LOM needs to be dotted line. Also, either of μLOM or μVeLOM needs to be a dotted line (by looking at (c), μVeLOM is a dotted line).

[1]: Hao Li, Zheng Xu, Gavin Taylor, Christoph Studer, Tom Goldstein, “Visualizing the Loss Landscape of Neural Nets,” NeurIPS 2018.

**Questions:**

•	Why did you select only MLP and Transformers (for language and vision) to show results on?

•	I am wondering how the loss landscapes of the approaches on ResNet and Plain ResNet would look.

---

> ### Author Response · Authors · 2024-11-22
> **General comments and replies to requested changes**
>
> Thank you for taking the time to review our paper. We are pleased to hear the reviewer believes that overall our proposed optimizers show improved performance and stability and that the generalization to longer training horizons is interesting.
>
> We now address the reviewer's concerns. Each bolded paragraph title refers to one or multiple concerns that have been grouped together. All line numbers refer to the originally submitted manuscript (not the updated version).
>
> **Results on  ResNets and Plain ResNets** The loss landscape is independent of the optimizer. It depends on the data, the loss function, and the optimizee’s (network being optimized) functional form/parameters. It is, nonetheless, interesting to assess the performance of our learned optimizers for plain and residual networks. In section G of the appendix, we report the performance of $\mu$LOs and SP LOs on plain and residual networks, finding that our $\mu$LOs perform well for both outperforming SP LOs.
>
> **The technical contribution is simple** While we agree that the change is simple, the strength of our approach, indeed, lies in its simplicity. Changing the optimizee’s parameterization is a small but non-trivial change relative to the existing meta-training framework in learned optimization: it enables substantial improvements to meta-generalization with no added computational cost to the learned optimizer and no restriction on the problems that can be considered. As such, it represents a net improvement over the existing meta-training framework in learned optimization [4.1,4.2,4.3].
>
> **Sub-optimal performance relative to VeLO-4000 Fig.4 (d) & (e)** As mentioned on L321-323 the tasks in Fig.4 (d) and (e) fall under the meta-training distribution of  VeLO-4000. That is, VeLO-4000 **was meta-trained on variants of these tasks** while $\mu$LO$_M$ and $\mu$VeLO$_M$ were **only meta-trained on MLPs**. When smaller width versions of the tasks are included in $\mu$LO meta-training (subfigures (a) and (b)), we observe that our optimizers outperform VeLO-4000. Note that due to the simplicity of our approach, there is no reason it would not be compatible with VeLO-scale meta-training.
>
>
> **Why study transformers and MLPs** We chose to study transformers and MLPs because MLPs allow reporting performance on the meta-training distribution and transformers are the main architecture used in practice today.
>
> ---
> **Local References**
>
>
> [3.1] [VeLO: Training Versatile Learned Optimizers by Scaling Up, Metz et al.]
>
> [3.2] [Tasks, stability, architecture, and compute: Training more effective learned optimizers, and using them to train themselves, Metz et al.]
>
> [3.3] [Practical Tradeoffs Between Memory, Compute, and Performance in Learned Optimizers, Metz et al., CoLLAs 2022]
>
> [3.4] [Yang et al., Tensor Programs V: Tuning Large Neural Networks via Zero-Shot Hyperparameter Transfer, Neurips 2021]

---

> ### Author Response · Authors · 2024-11-29
> **We are available to answer any remaining questions or concerns you may have**
>
> Dear reviewer umKW,
>
> Thank you for taking the time to review our paper.
>
> As the ICLR discussion period comes to an end, we would like to offer our assistance in responding to any remaining questions or concerns you may have about our paper.
>
> Best regards,
>
> The authors

---

> > ### Comment · Reviewer_umKW · 2024-11-30
> >
> > Thank the authors for their response. I went through the response, and confirmed my score.

---

### Official Review · Reviewer_s7Qy · 2024-11-03

**Soundness:** 3
**Presentation:** 2
**Contribution:** 3
**Rating:** 6
**Confidence:** 3

**Summary:**

The paper investigates the limitations of current learned optimizers (LOs)—small, data-driven neural networks designed to optimize other neural models—in their ability to generalize to longer optimization horizons and larger models than those encountered during meta-training. The authors derive the Maximal Update Parametrization (muP) for two popular LO architectures and propose a simple meta-training recipe for mu-parameterized learned optimizers (muLOs). Empirical evaluations demonstrate that muLOs can match or exceed the performance of VeLO—a state-of-the-art LO trained with 4000 TPU-months of compute—when applied to large-width models.

**Strengths:**

1. Effective Identification of Limitations in Learned Optimizers: The authors clearly pinpoint the generalization issues of current LOs, especially their generalization issues with larger models and longer training horizons.
2. Innovative use of Maximal Update Parametrization: By applying muP to LOs, they leverage theoretical guarantees for zero-shot hyperparameter tuning, going further to transfer optimizers.
3. Practical Solution to Costly Hyperparameter Optimization: The proposed muLOs offer a way to avoid expensive HPO on each new architecture and task, addressing a significant practical challenge.
4. Significance in the Field: By enhancing the generalization capabilities of LOs and making them more practical for large-scale applications.

**Weaknesses:**

1. Lack of Information on Computational Cost: The paper does not provide detailed information about the overall computational resources required for the method, spent on hyperparameter tuning for the mu-based methods. This is crucial for practitioners who may need to pre-train the optimizer themselves and assess the feasibility of implementing the method.
2. High Pre-Training Computational Cost: According to Line 356, the pre-training of an LO with already tuned hyperparameters amounts to 16 TPU-months. This is a substantial computational expense, prohibiting practitioners from pre-training an LO themselves. This means that a SoTA method would be preferred, and hence the comparison to VeLO seems more critical.
3. Unclear Method Overhead: There is insufficient information regarding the computational overhead introduced by using the LOs compared to standard optimizers. Specifically, the paper does not quantify how much slower the training process becomes when employing the neural network-based optimizer. For practical applications, understanding this overhead is essential for adjusting compute budgets and training timelines accordingly.
4. Limited Clarity in Results Presentation: Throughout the paper, the primary metric presented is the training loss. However, details about how this metric is calculated and presented are lacking. For instance, in Figure 2, it is unclear whether the loss is measured exactly at the presented iteration if the error bars represent standard deviation across different seeds, or how the results would differ if showing the best loss achieved so far or using a moving average.
5. Absence of downstream Evaluation Metrics.
6. Incomplete Presentation of Experimental Results: In Section 4.1.1, the authors mention collecting data for 1,000 training steps, but the plots presented display only 50% of this range (Figure 3 + appendix). Additionally, information about the number of seeds used and the variability across runs is missing.
7. Ambiguities in Generalization Evaluation (Figure 4):
* Convergence Similarities: In Figures 4a and 4b, the loss curves for muAdam and the proposed muLOs converge to similar values. This suggests that the primary benefit is improved early performance (anytime performance), but without considering the method's overhead, it's unclear how significant this benefit is in practical applications.
* Behavior of muAdam in Figure 4c: The loss trajectory of muAdam has an unexpected shape in Figure 4c without a clear explanation, making it difficult to understand the underlying reasons.
* Confusing Legend and Color Coding: The legend in Figure 4 uses the same markers for VeLO and LO, and for muVeLO and muLO, making it challenging to distinguish between the different methods.
8. Terminology Clarification Needed: The paper refers to muAdam, but it is unclear whether this refers to muAdam or muAdamW.
9. Inconsistent Performance Across Tasks (Figure 5): In Figure 5a, muLO appears to diverge, and across the tasks in Figures 5a and 5b, the ranking between muLO and muVeLO changes significantly. This inconsistency makes it difficult to determine which optimizer performs better overall and under what conditions, complicating the decision-making process for practitioners.
10. Questionable Claims About Stability: The claim in Line 516 that "muVeLO_M stably decrease training loss over time for each task" seems inaccurate. In Figure 6a, the loss begins to diverge, contradicting this statement. This raises concerns about the reliability of the method and the validity of the conclusions drawn.
11. High Variance in muVeLO Performance: The figures indicate that muVeLO exhibits a higher standard variance in its performance, while other methods show minimal variance. Understanding why muVeLO has higher variability is important for assessing its reliability and for guiding its practical adoption.

**Questions:**

1. Can you provide detailed information on the computational resources required for your method, specifically the hyperparameter tuning of the mu-based methods?
2. Given that retraining of both the LO and VeLO is infeasible, how do you justify LO use compared to the SoTA method?
3. How much additional computational overhead does using the LO introduce compared to standard optimizers/ SoTA, and how does this affect training speed?
4. Could you clarify how the training loss is calculated and presented throughout the paper, and whether the error bars in Figure 2 represent standard deviation across different seeds?
5. Could you include any validation metric, to assess the practical impact of your method?
6. In Figure 3, why do the plots display only 50% of the collected data, and could you provide information about the number of seeds used and variability across runs?
7. Is muAdam in your experiments referring to muAdam or muAdamW, and could you clarify this terminology?
8. Given the inconsistent performance between muLO and muVeLO across tasks (as seen in Figures 5a and 5b), can you provide guidance on which optimizer performs better under specific conditions?
9. How do you reconcile the claim that "muVeLO_M stably decreases training loss over time for each task" (Line 516) with the divergence observed in Figure 6a?
10. Why does muVeLO exhibit higher variance in its performance compared to other methods, and what are the implications for its practical adoption?

---

> ### Author Response · Authors · 2024-11-22
> **General comments and replies to requested changes**
>
> Thank you for taking the time to review our paper. We are pleased that reviewer s7Qy believes we effectively identify issues with currently learned optimizers, that our use of $\mu$P in this context is innovative, that $\mu$LOs offer a practical way to reduce HPO costs, and that our work can make LOs “more practical for large-scale applications”.
>
> We now address the reviewer's concerns. Each bolded paragraph title refers to one or multiple concerns that have been grouped together. All line numbers refer to the originally submitted manuscript (not the updated version).
>
> **Computational Cost of Meta-training is 16 TPU-months** False. The computational cost of meta-training our LOs is provided on line 19 of the abstract, line 356 puts it in the context of VeLO’s meta-training, and line 356 discusses it again. The value on line 356 is given as a percentage, therefore, this corresponds to 0.16 TPU months, less than 5 TPU-days.
>
> **Computational cost of $\mu$LOs** This is addressed in the general reply.
>
>
> **Figure 2 captions missing details** Thank you for pointing this out. We have updated the caption accordingly.
>
> **Regarding downstream evaluation** The main focus of our paper is to study the meta-generalization of $\mu$LOs applied to larger-width networks. Therefore, similar to prior published work [2.3], we choose to exclusively study training loss for simplicity.  Studying the generalization of the optimizee to downstream tasks is an important orthogonal direction that has been explored multiple times in previous works, as mentioned in L124-127. Specifically, it is well known that the following strategies can improve optimizee generalization: targeting validation loss during meta-training [2.7], using weight decay [2.6],  or using flatness-aware regularization [2.5].
>
>
> **“In Section 4.1.1, the authors mention collecting data for 1,000 training steps”** This is incorrect, line 375 (section 4.1.1) reports that these plots show 500 steps of training.
>
> **Performance relative to $\mu$Adam in Figure 4a and 4b and computational cost** Indeed, $\mu$ Adam (tuned for 500 trials) is a strong baseline. However, our goal is to improve meta-generalization to wider networks, a longstanding problem in learned optimization [2.1]. Therefore, it is most important to compare our approach to previous work in learned optimization, that is to the SP LOs and VeLO, which we handily beat in 4a and 4b. Note that our optimizers outperform$\mu$Adam in out-of-distribution tasks 4c, 4d, and 4e.
>
> **Behavior of muAdam in Figure 4c** $\mu$Adam is tuned for the ImageNet MLP width=1024 task (see Table 1). The results in Figure 4c demonstrate that $\mu$Adam’s hyperparameters aren't transferable to CIFAR-10 while $\mu$LO$_M$and $\mu$VeLO’s hyperparameters are. This shows the strength of our $\mu$LO framework which can achieve stronger meta-generalization than $\mu$-transfer [2.4].
>
> **No mention that  $\mu$Adam does not use weight decay** Thank you for suggesting this. We have updated section 4 to explicitly specify that no weight decay is used with $\mu$Adam.
>
> **Inconsistent best performer among $\mu$LO$_M$and $\mu$VeLO** This is addressed in the general reply.
>
>
> **$\mu$LO$_M$ diverges in Figure 5a** We disagree that $\mu$LO diverges in Figure 5a. We acknowledge that  $\mu$LO’s loss slightly increases after iteration 2000 but it is inconclusive whether it will diverge later on in training. It is important to compare $\mu$LO$_M$ to the LO$_M$ baseline in this figure. We note that $\mu$LO$_M$ reaches the final loss of value LO$_M$ after ~ 300 training iterations (16x speedup). Note that the SP baseline VeLO$_M$ (purple dashed line) does diverge in Figure 5a after the meta-training horizon (red dashed line). This demonstrates the improvements of  $\mu$LOs over previous work with respect to generalization to deeper networks.
>
> **$\mu$VeLO$_M$ begins to diverge in Figure 6a** $\mu$VeLO$_M$ does NOT diverge in Figure 6a. Perhaps you are referring to the SP baseline VeLO$_M$ (purple dashed line) which immediately diverges after the meta-training horizon (red dashed line) in this figure, demonstrating the improvements of  $\mu$LOs over previous work with respect to generalization to longer training horizons. We stand by our claim that "$\mu$VeLO$_M$ stably decrease training loss over time for each task" in Figure 6.
>
> **Variance of $\mu$VeLO** Answered in the general reply.

---

> ### Author Response · Authors · 2024-11-22
> **Local references for the author's reply**
>
> **Local References**
>
>
> [2.1] [VeLO: Training Versatile Learned Optimizers by Scaling Up, Metz et al.]
>
> [2.2] [Tasks, stability, architecture, and compute: Training more effective learned optimizers, and using them to train themselves, Metz et al.]
>
> [2.3] [Practical Tradeoffs Between Memory, Compute, and Performance in Learned Optimizers, Metz et al., CoLLAs 2022]
>
> [2.4] [Yang et al., Tensor Programs V: Tuning Large Neural Networks via Zero-Shot Hyperparameter Transfer, Neurips 2021]
>
> [2.5]  [Junjie Yang et al., Learning to Generalize Provably in Learning to Optimize, AISTATS 2023]
>
> [2.6] [Harrison et al. A Closer Look at Learned Optimization: Stability, Robustness, and Inductive Biases, Neurips 2022]
>
> [2.7] [Understanding and correcting pathologies in the training of learned optimizers, ICML 2019]

---

> ### Author Response · Authors · 2024-11-27
> **Reply to follow-up reservations**
>
> **Possible divergence in Figure 5a** After re-training the models in Figure 5, we noticed that the runs for $\mu$LO$_M$ in Fig. 5a (6 months old) were affected by a bug in the ViT’s $\mu$-parameterization that was fixed a while ago. We have updated the plot accordingly and verified that no other results are affected in the paper. The training curve now shows no sign of divergence. Thank you for asking about this.
>
> **$\mu$VeLO small loss increase in Figure 6a** The VeLO architecture takes as input the number of training steps remaining, thus, requiring the user to specify the total number of training steps (**total_steps**) a-priori (e.g. as is done for many LR schedules in practice). Therefore, at each step, VeLO’s LSTM is conditioned on an embedding that provides the number of training steps remaining, allowing it to learn a schedule. Previous work analyzing VeLO-4000’s behavior has noted that changing the value of the **total_steps** hyperparameter leads to variable performance [2.8]. Specifically, they found that increasing the value of **total_steps** does not always lead to better performance  [2.8]. To answer the reviewer’s question about extending training in Figure 6a, we have included training curves for $\mu$VeLO$_M$ on the width $1024$ ViT ImageNet task (same task as in 6a) in section H of the appendix. We show curves for $40,000$ steps of training and $25,000$ steps of training. We observe that $\mu$VeLO$_M$ can successfully optimize for $40,000$ training steps. However, we note this training curve underperforms $\mu$VeLO$_M$ using **total_steps**=25,000, similar to what was found in previous work for VeLO-4000 [2.8]. We note that this issue is specifically related to the VeLO architecture and not to $\mu$LOs
>
> **$\mu$Adam in Figure 5c** $\mu$Adam was tuned for 1000 training steps on a width 1024 MLP ImageNet task. Please note that it does not suffer from instability during the tuning window (first 1000 steps) in Figure 5c. While [2.4] shows that, in the cases they study, there is *empirical* generalization of $\mu$Adam’s hyperparameters to longer training unrolls, there is no reason to expect these findings hold for all tasks.  We observe that  $\mu$Adam’s hyperparameters are not transferable beyond the tuning horizon (1000 iterations)  to the Cifar10 task in Figure 5c. We have added a clarifying sentence in section 4.1.2 paragraph 4 stating that $\mu$Adam suffers from instability outside of its tuning window in figure 5c.
>
> **Random seeds used and performance variation across multiple runs** All of our plots reporting training loss show the average loss across 5 random seeds. The error bars in these plots report the standard error. Each seed corresponds to a different ordering of training data and a different initialization of the optimizee. We have added a sentence clarifying this in the introductory paragraph of section 4.1 and have made sure the captions of every figure throughout the entire paper information about seeds and error bars if there are any. For the performance variability, we assume the reviewer is referring to the variance in some $\mu$VeLO$_M$ training curves and its performance relative $\mu$LO$_M$. Please see the general comment for our answer to such concerns.
>
> Thank you for taking the time to thoroughly review our paper. Please let us know if you have any other questions or reservations.
>
> ---
> **Local references**
>
> [2.4] [Yang et al., Tensor Programs V: Tuning Large Neural Networks via Zero-Shot Hyperparameter Transfer, Neurips 2021]
>
> [2.8] [Is Scaling Learned Optimizers Worth It? Evaluating The Value of VeLO’s 4000 TPU Months]

---

### Official Review · Reviewer_i2Y6 · 2024-11-07

**Soundness:** 2
**Presentation:** 2
**Contribution:** 3
**Rating:** 5
**Confidence:** 4

**Summary:**

This paper proposes $\mu$LO, a new approach, inspired by hyperparameter transfer with the maximal update parameterization ($\mu$P), for training learned optimizers. Using $\mu$P for a model's initialization and weight updates has been shown to enable transferring high quality hyperparameter settings for the optimizer to networks of greater width, depth and longer training horizons.  Similarly, $\mu$LO addresses the limited ability of previous learned-optimizers to generalize to wider networks and longer training horizons by deriving the learned optimizer update required by $\mu$P.  Experiments show that $\mu$LO generalizes better than learned optimizers trained with standard parameterizations when applied to networks of larger width, depth, and longer training horizon as was observed for $\mu$P in the standard hyperparameter transfer settings.

**Strengths:**

- The experimental results are quite compelling and show $\mu$ LO addresses the generalization issues of previous incantations of learned optimizers.
- This is an interesting application of $\mu$ P and could potentially reduce the cost of model tuning dramatically.  However, the bar is quite high and will require additional work to compare $\mu$ LO with newer optimizers like Shampoo and Modula.

**Weaknesses:**

- While the experimental results are strong, they do not back up the claimed benefit that $\mu$ LO addresses the three issues of $\mu$ -transfer by avoiding the need for hyperparameter tuning for reach new architecture and task.
- The experimental results are missing comparisons to VeLO-4000 in Tables 5 and 6.  I think another reasonable baseline is per-task tuned Adam and $\mu$ Adam in the resource constrained setting to see if $\mu$ LO obvious the need for task specific optimizer tuning.
- The proposed $\mu$ LO does not take into account followup work on $\mu$ P that extends the original theoretical justification for transferring across width to depth (see [Tensor Program VI](https://arxiv.org/abs/2310.02244)).  Hence, the generalization across depth is more consequential then intended and may be improved with an explicit maximal update parameterization for depth.
- The legend for Figure 4 does not distinguish between dashed and solid lines.

**Questions:**

- According to Table 1, $\mu$ LO_M and $\mu$ VeLO_M are trained on ImageNet classification using 3-layer MLP with 3 different width.  Is that correct?  Does that mean transformer and ViT architectures studied in Figure 4 d and e were not seen during meta-training and are there to show generalization to unseen architectures?
- Please include VeLO-4000 in Figures 5 and 6.
- Why is $\mu$ LO better than $\mu$ VeLO on average despite VeLO being a more powerful model?
- How does $\mu$ LO compare to a per task tuned Adam/ $\mu$ Adam?
- What is the overhead of using the learned optimizer per gradient step relative to Adam?

---

> ### Author Response · Authors · 2024-11-22
> **General comments and replies to requested changes**
>
> Thank you for taking the time to review our paper. We are pleased that reviewer i2Y6 believes that $\mu$LO addresses generalization issues of previous work and that our approach could reduce the cost of model tuning dramatically.
>
> We now address the reviewer's concerns. Each bolded paragraph title refers to one or multiple concerns that have been grouped together. All line numbers refer to the originally submitted manuscript (not the updated version).
>
>
> **Overclaims in Section 3.3** We believe that the reviewer is referring to Section 3.3 Paragraph 1 and, in particular, to our suggestion that $\mu$LOs allow amortizing the tuning cost of multiple architectures, unlike $\mu$-transfer [1.4]. We understand your concern: while the experiments currently demonstrate strong generalization to unseen tasks and architecture families (e.g., as is the case for ViT, LM in Figure 4 (d) and (e)), the performance is not necessarily optimal in the sense that training an LO on small versions of unseen tasks/architectures may perform better than a model attempting to meta-generalize. Our goal for Section 3.3 Paragraph 1 was to state that, unlike $\mu$-transfer, the $\mu$LO framework allows meta-generalizing across unseen tasks/architecture families (a significant weakness of $\mu$-transfer). We have revised the text to clarify and tone down those claims in Sec. 3.3. Thank you for bringing this to our attention.
>
>
> **Regarding Figure 5 and 6 baselines** We emphasize that these figures study additional advantages of $\mu$LO for learned optimization beyond those it was designed to address (width scaling under a certain unroll length). Therefore, these figures only compare with the existing meta-training paradigm in learned optimization. In Figure 6, our focus is to establish the effect of $\mu$LO on generalization to longer unrolls. Thus, we directly compare to learned optimizers meta-trained identically (with the same unroll length) but using Standard Parameterization, which is the current paradigm in the learned optimization literature [1.1,1.2,1.3]. VeLO-4000 is meta-trained on a much longer unroll length (20k-200k steps) and as discussed in L321 is not a fair baseline, thus it is not only an unfair comparison but distracts from the question asked by Fig 6. For Figure 5, similarly, our goal is to directly show the unexpected advantage of $\mu$LO on generalization to deeper networks over learned optimizers meta-trained under the existing paradigm.
>
> **Regarding per-task tuned baselines** We did not include the per-tasked re-turned $\mu$Adam baseline in our main results because our $\mu$LOs do not benefit from meta-training on these tasks and the comparison would not be fair. We do agree that this is an interesting baseline to consider, however. In section F of the appendix, we include results on the ViT w=3072 task where $\mu$Adam was tuned for 500 trials in the resource-constrained setting (e.g. on a smaller width version of the task) as you suggested. The results show that the re-tuned $\mu$Adam improves over its predecessor, but is outperformed by $\mu$LO$_M$ and $\mu$VeLO.
>
> **$\mu$ Depth** Thank you for bringing this up. Depth $\mu$P only works for residual networks with a block depth of 1. Since most networks used in practice today (e.g., the transformer) have a block depth > 1 they are incompatible with $\mu$Depth. We, therefore, chose not to study meta-generalization under $\mu$Depth. The main focus of our paper is to study the meta-generalization of $\mu$LOs applied to larger-width networks. However, we noticed that this choice of parameterization is beneficial for meta-generalization to deeper networks and longer unrolls which is very interesting to the learned optimization community, so we also included results for these cases. We have added a brief discussion of \muDepth and our choice not to study \mu depth in section 2, paragraph 3.
>
> **Question regarding the meta-training of $\mu$LO$_M$ and $\mu$VeLO$_M$** Yes, the learned optimizers were only meta-trained on the tasks shown in Table 4 for $1000$ steps. This means the tasks seen in Figure 4 (d) and (e) were NOT seen during meta-training and that these subfigures show meta-generalization to new tasks. Moreover, all figures in the paper showing training beyond $1000$ iterations are out-of-distribution with respect to training duration for our $\mu$LO$_M$ and $\mu$VeLO$_M$.
>
>
> **Why is LO better than VeLO on average despite VeLO being a more powerful model?** This question is addressed in the general reply.
>
>
> **Overhead of $\mu$LO$_M$ and $\mu$VeLO$_M$** This is addressed in the general reply.

---

> > ### Comment · Reviewer_i2Y6 · 2024-11-26
> > **Post author response**
> >
> > Thank you for responding to the issues I brought up.  However, I do not feel like the main weaknesses I raised were sufficiently addressed due to lack of additional experiments for baselines or incorporation of $\mu$ depth.  Therefore, I will maintain my score.

---

> > > ### Author Response · Authors · 2024-12-02
> > > **We have run additional experiments incorporating Depth$\mu$P**
> > >
> > > As requested by the reviewer we have now run some additional experiments with Depth-$\mu$P.
> > >
> > >
> > > We first want to re-iterate that Tensor Programs VI [1.5] shows theoretically that Depth$\mu$P admits hyperparameter transfer in “resnets where each block has only one layer” [1.5] and **only** in this case. The authors illustrate both theoretically and empirically that “if each block is deeper (such as modern transformers), then [they] find fundamental limitations in all possible infinite-depth limits of such parametrizations”[1.5]. Specifically, in section 9 of [1.5], the authors show an impossibility result **for block depth >=2**, demonstrating that **this case cannot admit hyperparameter transfer** under the Depth-$\mu$P parameterization they propose. Thus, even when parameterizing $\mu$LOs in Depth-$\mu$P, any findings for generalization to deeper counterparts of networks with Block depth >=2 (such as modern transformers) **will still be more consequential than intended**. Therefore, Depth-$\mu$P can only help address the weakness that reviewer 12Y6 originally mentioned for the block depth=1 case.
> > >
> > > Since Depth-$\mu$P is only valid for residual networks with block size=1, we evaluate Depth-$\mu$LOs for residual networks with a block depth of 1. Our new experiments incorporate one new baseline, Depth-$\mu$Adam, and compare it to $\mu$LO$_M$ paramterized in Depth-$\mu$P, henceforth referred to as Depth-$\mu$LO$_M$.
> > >
> > > **Depth-$\mu$Adam** Depth-$\mu$Adam is tuned on a with=128 and depth=16 residual MLP ImageNet-32x32x3 classification task. In addition to sweeping the same tunable multipliers used for $\mu$Adam, Depth-$\mu$Adam also tunes *depth_LR_multiplier* and *depth_branch_multipler* by sweeping values in {$\{2^{−4} , 2^{−2} , 1, 2^2 , 2^4\}$}. This results in 12500 total hyperparameter configurations.
> > >
> > >
> > > Please note that the total GPU-time for Depth-$\mu$Adam’s hyperparameter sweep now matches the meta-training time of the Depth-$\mu$LO$_M$. In the Table below, we report the final loss for with=128,depth=64 (4x tuning) and with=128,depth=128 MLP tasks (8x tuning). We observe that all optimizers following a  Depth-$\mu$-parameterization improve the loss as the depth increases. In contrast, SP LO diverges. Comparing within optimizers in Depth-$\mu$P, we observe that Depth-$\mu$LO$_M$ outperforms Depth-$\mu$Adam. These results are consistent with what was observed throughout our paper for width and they illustrate, as the reviewer suggested, that applying Depth-$\mu$P can help improve generalization to deeper networks. However, we note that **this parameterization only admits hyperparameter transfer for networks with a block depth=1** [1.5]. While it is outside the scope of our work, we believe that it will be very important for future work to develop parameterizations transferrable for architecutres using a block depth>=2 (e.g., transformers). Thank you for suggesting we explore this, we will incorporate these results in a new section of the appendix.
> > >
> > > Generalization to deeper tasks in Depth-$\mu$P.
> > > | Task               | Depth=64 Residual MLP | Depth=128 Residual MLP |
> > > |---------------------|-----------------------|-------------------------|
> > > | SP LO              | Diverges              | Diverges               |
> > > | Depth-$\mu$LO$_M$  | **5.26**                  | **5.23**                   |
> > > | Depth-$\mu$Adam    | 5.29                  | 5.27                   |
> > >
> > >
> > > ---
> > > **Local references**
> > >
> > > [1.5] [Tensor Programs VI: Feature Learning in Infinite Depth Neural Networks]

---

> ### Author Response · Authors · 2024-11-22
> **Local references for the author's reply**
>
> **Local References**
>
> [1.1] [VeLO: Training Versatile Learned Optimizers by Scaling Up, Metz et al.]
>
> [1.2] [Tasks, stability, architecture, and compute: Training more effective learned optimizers, and using them to train themselves, Metz et al.]
>
> [1.3] [Practical Tradeoffs Between Memory, Compute, and Performance in Learned Optimizers, Metz et al., CoLLAs 2022]
>
> [1.4] [Yang et al., Tensor Programs V: Tuning Large Neural Networks via Zero-Shot Hyperparameter Transfer, Neurips 2021]

---

### Author Response · Authors · 2024-11-22
**General reply to all reviewers**

**General comment:**  We would like to thank all reviewers for their thoughtful insights and constructive criticism about our paper. We are pleased that reviewers believe we effectively highlight generalization issues with current learned optimizers [s7Qy] and that $\mu$ learned optimization ($\mu$LO) addresses them [i2Y6,oHHy]. We are particularly pleased that s7Qy thinks the use of $\mu$P in this context is innovative and that $\mu$LOs offer a practical way to dramatically reduce model tuning cost [i2Y6, s7Qy]. Finally, we are happy that umKW believes our method improves overall performance and oHHy finds the empirical performance to be impressive.

**Computational cost of $\mu$LO$_M$and $\mu$VeLO$_M$** We chose not to include any such comparison in our paper as the main focus of our work is to study the meta-generalization of $\mu$LOs applied to larger-width networks (orthogonal to improving learned optimizer computational cost). As mentioned on L538, the benefits of $\mu$LOs come at zero added computational cost relative to learned optimizers from previous work. Nonetheless, we have included results in the table below that compare the computational cost of the learned optimizers in this study to that of AdamW. We refer reviewers to [1] (section B.7) and [3] (sections 4.1,4.2,4.3,4.4) for a more detailed account of the computational cost of $\mu$VeLO$_M$ and $\mu$LO$_M$, respectively.


*Table 1 Computational cost of the learned optimizers in our study and AdamW* We report the forward and backward pass time of the optimizee, the optimizer step time, and the total step time per iteration when training ResNet18 on ImageNet. Each number is the median step time over the last 250 training iterations. All timings were measured on an L40S Nvidia GPU.


| Optimizer         | Total Step Time | Optimizer Step Time | FwBw Time |
|-------------------|-----------------|---------------------|-----------------|
| AdamW            | 0.0199         | 0.0022              | 0.0177          |
| $\mu$LO$_M$      | 0.0219         | 0.0039              | 0.0180          |
| $\mu$VeLO$_M$    | 0.0253         | 0.0076              | 0.0177          |
| LO$_M$           | 0.0220         | 0.0041              | 0.0179          |
| VeLO$_M$         | 0.0255         | 0.0077              | 0.0178          |


**$\mu$VeLO$_M$ variance and performance relative to $\mu$LO**  $\mu$VeLO$_M$ uses an LSTM to generate the parameters of a small per-parameter MLP for each tensor in the optimizee. This two-level structure substantially increases its capacity relative to $\mu$LO$_M$(only a  single per-parameter MLP). While $\mu$VeLO$_M$ achieves generalization to wider networks, it appears to suffer from greater variance than $\mu$LO$_M$, which is not observed in VeLO-4000 (trained on a large distribution of tasks).  Moreover, $\mu$LO$_M$ outperforms $\mu$VeLO$_M$ on most tasks, despite its lower capacity. We hypothesize $\mu$VeLO$_M$ may require more extensive meta-training beyond the scope of this research to see it outperform $\mu$LO$_M$. Our goal here was simply to demonstrate that our parameterization also works for the VeLO architecture.


**Figure 4 legend is incorrect** We have fixed this in the updated version. Thank you all for pointing this out.


---

**Local References**

[1] [VeLO: Training Versatile Learned Optimizers by Scaling Up, Metz et al.]

[2] [Tasks, stability, architecture, and compute: Training more effective learned optimizers, and using them to train themselves, Metz et al.]

[3] [Practical Tradeoffs Between Memory, Compute, and Performance in Learned Optimizers, Metz et al., CoLLAs 2022]

[4] [Yang et al., Tensor Programs V: Tuning Large Neural Networks via Zero-Shot Hyperparameter Transfer, Neurips 2021]

---

### Author Response · Authors · 2024-11-22
**Changes to the manuscript**

**Here is a list of our changes to the manuscript based on reviewer feedback.**

- We have revised the text to clarify and tone claims in Sec. 3.3 with respect to i2Y6’s feedback.
- We have added a brief discussion of Depth-$\mu$P and our choice not to study Depth-$\mu$P in section 2, paragraph 3. This change was made due to i2Y6's feedback.
- In section F of the appendix, we have added a comparison of the performance of our $\mu$LOs to $\mu$Adam tuned for 500 trials on a ViT task. This change was made due to i2Y6's feedback.
- We have added missing details from Figure 2 caption with respect to s7Qy’s feedback.
- We have added an evaluation of ResNets in section G of the appendix concerning reviewer umKW’s feedback.
- We have updated the manuscript by adding a paragraph to section 3.2 that adds more background about $\mu$P to the text in accordance with oHHy’s feedback.
- We have minimally updated the abstract, contributions, and section 4 introductory paragraph to emphasize that the findings of improved generalization to deeper networks and more training steps are purely empirical and not theoretically motivated.

---

### Author Response · Authors · 2024-11-27
**Changes to the manuscript II**

**Here is a list of our newest changes to the manuscript based on reviewer feedback.**

- We have updated Figure 5a. While responding to reviewer s7Qy, we noticed that the runs for $\mu$LO$_M$ in Fig. 5a (6 months old) were affected by a bug in the ViT’s $\mu$-parameterization that was fixed a while ago. We have updated the plot with corrected runs and verified that no other results are affected in the paper. $\mu$LO$_M$ now consistently decreases the loss throughout training in Fig. 5a.

- We have added section H to the appendix of our paper, showing training curves of $\mu$VeLO$_M$ for 40,000 training steps.

- We have made sure that there is a sentence in the captions of every figure stating the number of trials used for each curve and explaining what the error bars designate.

- We have added a clarifying sentence to section 4.1.2 paragraph 4 to explain the instabilities of $\mu$Adam in figure 5c.

---

### Author Response · Authors · 2024-11-27
**Changes to the manuscript III**

**Here is a list of our latest changes to the manuscript based on reviewer feedback.**
- We have enhanced the presentation of Figures 2,11,12,13,14 by replacing the legends in each figure with a global color bar and removing redundant x and y labels.

---

### Meta-Review · Area_Chair_3Pqp · 2024-12-23

**Metareview:**

The paper introduces µ-learned optimizers (µLOs), leveraging Maximal Update Parametrization (µP) to enhance generalization to larger models, deeper networks, and longer training horizons. µLOs achieve competitive or superior performance to state-of-the-art methods like VeLO, with significantly lower computational costs, effectively addressing challenges in hyperparameter tuning. Strenths of the paper include computational efficiency and performance of the approach in the benchmarks studied. However, concerns were raised about the limited novelty of the contribution, as it primarily applies µP to learned optimizers. Also, some key baselines, such as VeLO-4000 and per-task tuned optimizers, were not fully evaluated. Performance inconsistencies on certain out-of-distribution tasks were also pointed out. The main reason for my recommendation to reject is the incomplete experimental comparison.

**Additional Comments On Reviewer Discussion:**

The reviewers made various changes during the rebuttal, including bug fixes, new experiments, and extensive clarifications in text and visuals. However, novelty and incomplete baselines remained as limitations.

---

### Decision · Program_Chairs · 2025-01-22

Reject